# Transformers Learn Higher-Order Optimization Methods for In-Context Learning: A Study with Linear Models

## Abstract

Transformers are remarkably good at *in-context learning* (ICL)—learning from demonstrations without parameter updates—but how they perform ICL remains a mystery. Recent work suggests that Transformers may learn in-context by internally running Gradient Descent (GD), a first-order optimization method. In this paper, we instead demonstrate that Transformers learn to implement higher-order optimization methods to perform ICL. Focusing on in-context linear regression, we show that Transformers learn to implement an algorithm very similar to *Iterative Newton's Method*, a higher-order optimization method, rather than Gradient Descent. Empirically, we show that predictions from successive Transformer layers closely match different iterations of Newton's Method *linearly*, with each middle layer roughly computing 3 iterations. In contrast, *exponentially* more GD steps are needed to match an additional Transformers layer; this suggests that Transformers have an comparable rate of convergence with high-order methods, which are exponentially faster than GD. We also show that Transformers can learn in-context on ill-conditioned data, a setting where Gradient Descent struggles but Iterative Newton succeeds. Finally, we show theoretical results which support our empirical findings and have a close correspondence with them: we prove that Transformers can implement $k$ iterations of Newton's method with $\mathcal{O}(k)$ layers.

## 1 Introduction

Transformer neural networks (Vaswani et al., 2017) have become the default architecture for natural language processing (Devlin et al., 2019; Brown et al., 2020; OpenAI, 2023), and have even been adopted by other areas like computer vision (Dosovitskiy et al., 2021). As first demonstrated by GPT-3 (Brown et al., 2020), Transformers excel at *in-context learning* (ICL)—learning from input-output pairs provided as inputs to the model, without updating their model parameters. Through in-context learning, Transformer-based Large Language Models (LLMs) can achieve state-of-the-art few-shot performance across a wide variety of downstream tasks (Rae et al., 2022; Smith et al., 2022; Thoppilan et al., 2022; Chowdhery et al., 2022).

Given the importance of Transformers and ICL, many prior efforts have attempted to understand how Transformers perform in-context learning. Prior work suggests Transformers can approximate linear functions well in-context (Garg et al., 2022). Specifically to linear regression tasks, prior work has tried to understand the ICL mechanism and the dominant hypothesis is that Transformers learn in-context by running optimizations internally through gradient-based algorithms (von Oswald et al., 2022; 2023; Ahn et al., 2023; Dai et al., 2023).

This paper presents strong evidence for a competing hypothesis: Transformers trained to perform in-context linear regression learn to implement a higher-order optimization method rather than a first-order method like Gradient Descent. In particular, Transformers implement a method very similar to Newton-Schulz's Method, also known as the *Iterative Newton's Method*, which iteratively improves an estimate of the inverse of the design matrix to compute the optimal weight vector. Across many layers of the Transformer, subsequent layers approximately compute more and more iterations of Newton's Method, with increasingly better predictions; both eventually converge to the optimal minimum-norm solution found by ordinary least squares (OLS). Interestingly, this mechanism is

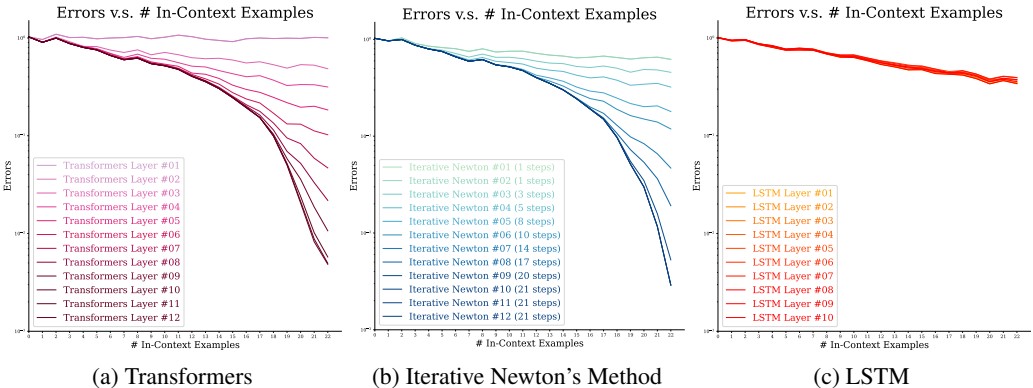

(a) Transformers  (b) Iterative Newton's Method  (c) LSTM

Figure 1: **Progression of Algorithms.** (a) Transformer's performance improves over the layer index $\ell$. (b) Iterative Newton's performance improves over the number of iterations $k$, in a way that closely resembles the Transformer. We plot the best-matching $k$ to Transformer's $\ell$ following Definition 4. (c) In contrast, LSTM's performance does not improve from layer to layer.

specific to Transformers: LSTMs do not learn these same higher-order methods, as their predictions do not even improve across layers.

We present both empirical and theoretical evidence for our claims. Empirically, Transformer induced weights and residuals are similar to Iterative Newton and deeper layers match Newton with more iterations (see Figures 1 and 9). Transformers can also handle ill-conditioned problems without requiring significantly more layers, where GD would suffer from slow convergence but Iterative Newton would not. Crucially, Transformers share the same rate of convergence as Iterative Newton and are exponentially faster than GD. Theoretically, we show that Transformer circuits can efficiently implement Iterative Newton, with the number of layers depending linearly on the number of iterations and the dimensionality of the hidden states depending linearly on the dimensionality of the data. Overall, our work provides a mechanistic account of how Transformers perform in-context learning that not only explains model behavior better than previous hypotheses, but also hints at what makes Transformers so well-suited for ICL compared with other neural architectures.

## 2 RELATED WORK

**In-context learning by large language models.** GPT-3 (Brown et al., 2020) first showed that Transformer-based large language models can "learn" to perform new tasks from in-context demonstrations (i.e., input-output pairs). Since then, a large body of work in NLP has studied in-context learning, for instance by understanding how the choice and order of demonstrations affects results (Lu et al., 2022; Liu et al., 2022; Rubin et al., 2022; Su et al., 2023; Chang & Jia, 2023; Nguyen & Wong, 2023), studying the effect of label noise (Min et al., 2022c; Yoo et al., 2022; Wei et al., 2023), and proposing methods to improve ICL accuracy (Zhao et al., 2021; Min et al., 2022a;b).

**In-context learning beyond natural language.** Inspired by the phenomenon of ICL by large language models, subsequent work has studied how Transformers learn in-context beyond NLP tasks. Garg et al. (2022) first investigated Transformers' ICL abilities for various classical machine learning problems, including linear regression. We largely adopt their linear regression setup in this work. Li et al. (2023) formalize in-context learning as an algorithm learning problem where a Transformer model implicitly constructs a hypothesis function at inference-time and obtain generalization bounds for ICL. Han et al. (2023) suggests that Transformers learn in-context by performing Bayesian inference on prompts, which can be asymptotically interpreted as kernel regression. Tarzanagh et al. (2023a) and Tarzanagh et al. (2023b) show that Transformers can find max-margin solutions for classification tasks and act as support vector machines. Zhang et al. (2023) prove that a linear attention Transformer trained by gradient flow can indeed in-context learn class of linear models. Raventós et al. (2023) explore how diverse pretraining data can enable models to perform ICL on new tasks.

**Do Transformers implement Gradient Descent?** A growing body of work has suggested that Transformers learn in-context by implementing gradient descent within their internal representations. Akyürek et al. (2022) summarize operations that Transformers can implement, such as multi-

plication and affine transformations, and show that Transformers can implement gradient descent for linear regression using these operations. Concurrently, von Oswald et al. (2022) argue that Transformers learn in-context via gradient descent, where one layer performs one gradient update. In subsequent work, von Oswald et al. (2023) further argue that Transformers are strongly biased towards learning to implement gradient-based optimization routines. Ahn et al. (2023) extend the work of von Oswald et al. (2022) by showing Transformers can learn to implement preconditioned Gradient Descent, where the pre-conditioner can adapt to the data. Bai et al. (2023) provides detailed constructions for how Transformers can implement a range of learning algorithms via gradient descent. Finally, Dai et al. (2023) conduct experiments on NLP tasks and conclude that Transformer-based language models performing ICL behave similarly to models fine-tuned via gradient descent. In this paper, we argue that Transformers actually learn to perform in-context learning by implementing a higher-order optimization method, not gradient descent. Predictions made by different Transformer layers match iterations of higher-order optimization methods better than they match iterations of gradient descent; moreover, Transformers can handle ill-conditioned data, unlike Gradient Descent.

**Mechanistic interpretability for Transformers.** Our work attempts to understand the mechanism through which Transformers perform in-context learning. Prior work has studied other aspects of Transformers' internal mechanisms, including reverse-engineering language models (Wang et al., 2022), the grokking phenomenon (Power et al., 2022; Nanda et al., 2023), manipulating attention maps (Hassid et al., 2022), and automated circuit finding (Conmy et al., 2023).

## 3 PROBLEM SETUP

In this paper, we focus on the following linear regression task. The task involves $n$ examples $\{\boldsymbol{x}_i, y_i\}_{i=1}^n$ where $\boldsymbol{x}_i \in \mathbb{R}^d$ and $y_i \in \mathbb{R}$. The examples are generated from the following data generating distribution $P_{\mathcal{D}}$, parameterized by a distribution $\mathcal{D}$ over $(d \times d)$ positive semi-definite matrices. For each sequence of $n$ in-context examples, we first sample a ground-truth weight vector $\boldsymbol{w}^\star \overset{\text{i.i.d.}}{\sim} \mathcal{N}(\boldsymbol{0}, \boldsymbol{I}) \in \mathbb{R}^d$ and a matrix $\boldsymbol{\Sigma} \overset{\text{i.i.d.}}{\sim} \mathcal{D}$. For $i \in [n]$, we sample each $\boldsymbol{x}_i \overset{\text{i.i.d.}}{\sim} \mathcal{N}(\boldsymbol{0}, \boldsymbol{\Sigma})$. The label $y_i$ for each $\boldsymbol{x}_i$ is given by $y_i = \boldsymbol{w}^{\star\top} \boldsymbol{x}_i$. Note that for much of our experiments $\mathcal{D}$ is only supported on

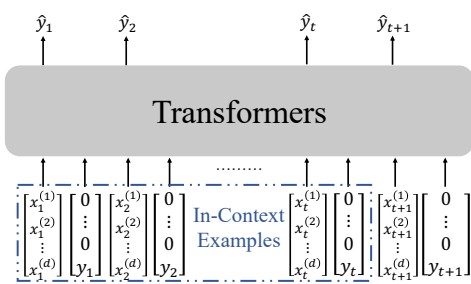

Figure 2: Illustration of how Transformers are trained to do in-context linear regression.

the identity matrix $\boldsymbol{I} \in \mathbb{R}^{d \times d}$ and hence $\boldsymbol{\Sigma} = \boldsymbol{I}$, but we also consider some distributions over ill-conditioned matrices which will give rise to ill-conditioned regression problems.

### 3.1 SOLVING LINEAR REGRESSION WITH TRANSFORMERS

We will use neural network models such as Transformers to solve this linear regression task. As shown in Figure 2, at time step $t + 1$, the model sees the first $t$ in-context examples $\{\boldsymbol{x}_i, y_i\}_{i=1}^t$, and then makes predictions for $\boldsymbol{x}_{t+1}$, whose label $y_{t+1}$ is not observed by the Transformers model.

We randomly initialize our models and then train them on the linear regression task to make predictions for every number of in-context examples $t$, where $t \in [n]$. Training and test data are both drawn from $P_{\mathcal{D}}$. To make the input prompts contain both $\boldsymbol{x}_i$ and $y_i$, we follow same the setup as Garg et al. (2022)'s to zero-pad $y_i$'s, and use the same decoder-only GPT-2 model (Radford et al., 2019) with softmax activation and causal attention mask (discussed later at definition 1).

We now present the key mathematical details for the Transformer architecture, and how they can be used for in-context learning. First, the causal attention mask enforces that attention heads can only attend to hidden states of previous time steps, and is defined as follows.

**Definition 1** (Causal Attention Layer). *An **causal** attention layer with $M$ heads and activation function $\sigma$ is denoted as* Attn *on any input sequence* $\boldsymbol{H} = [\boldsymbol{h}_1, \cdots, \boldsymbol{h}_N] \in \mathbb{R}^{D \times N}$, *where $D$ is the dimension of hidden states and $N$ is the sequence length. In the vector form,*

$$\tilde{\boldsymbol{h}}_t = [\text{Attn}(\boldsymbol{H})]_t = \boldsymbol{h}_t + \sum_{m=1}^M \sum_{j=1}^t \sigma\left(\langle \boldsymbol{Q}_m \boldsymbol{h}_t, \boldsymbol{K}_m \boldsymbol{h}_j \rangle\right) \cdot \boldsymbol{V}_m \boldsymbol{h}_j. \tag{1}$$

Vaswani et al. (2017) originally proposed the Transformer architecture with the Softmax activation function for the attention layers. Later works have found that replacing $\mathrm{Softmax}(\cdot)$ with $\frac{1}{t}\mathrm{ReLU}(\cdot)$ does not hurt model performance (Cai et al., 2022; Shen et al., 2023; Wortsman et al., 2023). The Transformers architecture is defined by putting together attention layers with feed forward layers:

**Definition 2** (Transformers). *An $L$-layer decoder-based transformer with Causal Attention Layers is denoted as* $\mathrm{TF}_{\boldsymbol{\theta}}$ *and is a composition of a MLP Layer (with a skip connection) and a Causal Attention Layers. For input sequence* $\boldsymbol{H}^{(0)}$*, the transformers $\ell$-th hidden layer is given by*

$$\mathrm{TF}_{\boldsymbol{\theta}}^{\ell}(\boldsymbol{H}^{(0)}) := \boldsymbol{H}^{(\ell)} = \mathrm{MLP}_{\boldsymbol{\theta}_{\mathrm{mlp}}^{(\ell)}}\left(\mathrm{Attn}_{\boldsymbol{\theta}_{\mathrm{attn}}^{(\ell)}}\left(\boldsymbol{H}^{(\ell-1)}\right)\right) \tag{2}$$

*where* $\boldsymbol{\theta} = \{\boldsymbol{\theta}_{\mathrm{mlp}}^{(\ell)}, \boldsymbol{\theta}_{\mathrm{attn}}^{(\ell)}\}_{\ell=1}^{L}$ *and* $\boldsymbol{\theta}_{\mathrm{attn}}^{(\ell)} = \{\boldsymbol{Q}_m^{(\ell)}, \boldsymbol{K}_m^{(\ell)}, \boldsymbol{V}_m^{(\ell)}\}_{m=1}^{M}$ *consists of $M$ heads at layer $\ell$.*

In particular for the linear regression task, Transformers perform in-context learning as follows

**Definition 3** (Transformers for Linear Regression). *Given in-context examples* $\{\boldsymbol{x}_1, y_1, \ldots, \boldsymbol{x}_t, y_t\}$*, Transformers make predictions on a query example* $\boldsymbol{x}_{t+1}$ *through a readout layer parameterized as* $\boldsymbol{\theta}_{\mathrm{readout}} = \{\boldsymbol{u}, v\}$*, and the prediction* $\hat{y}_{t+1}^{\mathrm{TF}}$ *is given by*

$$\hat{y}_{t+1}^{\mathrm{TF}} := \mathrm{ReadOut}\Big[\underbrace{\mathrm{TF}_{\boldsymbol{\theta}}^{L}(\{\boldsymbol{x}_1, \boldsymbol{y}_1, \cdots, \boldsymbol{x}_t, \boldsymbol{y}_t, \boldsymbol{x}_{t+1}\})}_{\boldsymbol{H}^{(L)}}\Big] = \boldsymbol{u}^{\top}\boldsymbol{H}_{:,2t+1}^{(L)} + v. \tag{3}$$

### 3.2 STANDARD METHODS FOR SOLVING LINEAR REGRESSION

Our central research question is:

> ***Does the algorithm Transformers learn for linear regression resemble any known algorithm?***

Here we discuss various known algorithms we compare Transformers with.

For any time step $t$, let $\boldsymbol{X}^{(t)} = [\boldsymbol{x}_1 \ \cdots \ \boldsymbol{x}_t]^{\top}$ be the data matrix and $\boldsymbol{y}^{(t)} = [y_1 \ \cdots \ y_t]^{\top}$ be the labels for all the datapoints seen so far. Note that since $t$ can be smaller than the data dimension $d$, $\boldsymbol{X}^{(t)}$ can be singular. We now consider various algorithms for making predictions for $\boldsymbol{x}_{t+1}$ based on $\boldsymbol{X}^{(t)}$ and $\boldsymbol{y}^{(t)}$. When it is clear from context, we will drop the superscript and refer to $\boldsymbol{X}^{(t)}$ and $\boldsymbol{y}^{(t)}$ as $\boldsymbol{X}$ and $\boldsymbol{y}$, where $\boldsymbol{X}$ and $\boldsymbol{y}$ correspond to all the datapoints seen so far.

**Ordinary Least Squares.** This method finds the minimum-norm solution to the objective:

$$\mathcal{L}(\boldsymbol{w} \mid \boldsymbol{X}, \boldsymbol{y}) = \frac{1}{2n}\|\boldsymbol{y} - \boldsymbol{X}\boldsymbol{w}\|_2^2. \tag{4}$$

The Ordinary Least Squares (OLS) solution has a closed form given by the Normal Equations:

$$\hat{\boldsymbol{w}}^{\mathrm{OLS}} = (\boldsymbol{X}^{\top}\boldsymbol{X})^{\dagger}\boldsymbol{X}^{\top}\boldsymbol{y} \tag{5}$$

where $\boldsymbol{S} := \boldsymbol{X}^{\top}\boldsymbol{X}$ and $\boldsymbol{S}^{\dagger}$ is the pseudo-inverse (Moore, 1920) of $\boldsymbol{S}$.

**Gradient Descent.** Gradient descent (GD) finds the weight vector $\hat{\boldsymbol{w}}^{\mathrm{GD}}$ by initializing $\hat{\boldsymbol{w}}_0^{\mathrm{GD}} = \boldsymbol{0}$ and using the iterative update rule:

$$\hat{\boldsymbol{w}}_{k+1}^{\mathrm{GD}} = \hat{\boldsymbol{w}}_k^{\mathrm{GD}} - \eta\nabla_{\boldsymbol{w}}\mathcal{L}(\hat{\boldsymbol{w}}_k^{\mathrm{GD}} \mid \boldsymbol{X}, \boldsymbol{y}). \tag{6}$$

It is known that Gradient Descent requires $\mathcal{O}\left(\kappa(\boldsymbol{S})\log(1/\epsilon)\right)$ steps to converge to an $\epsilon$ error where $\kappa(\boldsymbol{S}) = \frac{\lambda_{\max}(\boldsymbol{S})}{\lambda_{\min}(\boldsymbol{S})}$ is the *condition number*. Thus, when $\kappa(\boldsymbol{S})$ is large, Gradient Descent converges slowly (Boyd & Vandenberghe, 2004).

**Online Gradient Descent.** While GD computes the gradient with respect to the full data matrix $\boldsymbol{X}$ at each iteration, Online Gradient Descent (OGD) is an online algorithm that only computes gradients on the newly received data point $\{\boldsymbol{x}_k, y_k\}$ at step $k$:

$$\hat{\boldsymbol{w}}_{k+1}^{\mathrm{OGD}} = \hat{\boldsymbol{w}}_k^{\mathrm{OGD}} - \eta_k\nabla_{\boldsymbol{w}}\mathcal{L}(\hat{\boldsymbol{w}}_k^{\mathrm{OGD}} \mid \boldsymbol{x}_k, y_k). \tag{7}$$

Picking $\eta_k = \frac{1}{\|\boldsymbol{x}_k\|_2^2}$ ensures that the new weight vector $\hat{\boldsymbol{w}}_{k+1}^{\mathrm{OGD}}$ makes zero error on $\{\boldsymbol{x}_k, y_k\}$.

**Iterative Newton's Method.** This method finds the weight vector $\hat{\boldsymbol{w}}^{\mathrm{Newton}}$ by iteratively apply Newton's method to finding the pseudo inverse of $\boldsymbol{S} = \boldsymbol{X}^\top \boldsymbol{X}$ (Schulz, 1933; Ben-Israel, 1965).

$$
\begin{aligned}
\boldsymbol{M}_0 = \alpha \boldsymbol{S}, \text{ where } \alpha = \frac{2}{\|\boldsymbol{S}\boldsymbol{S}^\top\|_2}, &\qquad \hat{\boldsymbol{w}}_0^{\mathrm{Newton}} = \boldsymbol{M}_0 \boldsymbol{X}^\top \boldsymbol{y}, \\
\boldsymbol{M}_{k+1} = 2\boldsymbol{M}_k - \boldsymbol{M}_k \boldsymbol{S} \boldsymbol{M}_k, &\qquad \hat{\boldsymbol{w}}_{k+1}^{\mathrm{Newton}} = \boldsymbol{M}_{k+1} \boldsymbol{X}^\top \boldsymbol{y}.
\end{aligned}
\tag{8}
$$

This computes an approximation of the psuedo inverse using the moments of $\boldsymbol{S}$. In contrast to GD, the Iterative Newton's method only requires $\mathcal{O}(\log \kappa(\boldsymbol{S}) + \log\log(1/\epsilon))$ steps to converge to an $\epsilon$ error (Soderstrom & Stewart, 1974; Pan & Schreiber, 1991). Note that this is exponentially faster than the convergence rate of Gradient Descent.

### 3.3 LSTM

While our primary goal is to analyze Transformers, we also consider the LSTM architecture (Hochreiter & Schmidhuber, 1997) to understand whether Transformers learn different algorithms than other neural sequence models trained to do linear regression. In particular, we train a unidirectional $L$-layer LSTM, which generates a sequence of hidden states $\boldsymbol{H}^{(\ell)}$ for each layer $\ell$, similarly to an $L$-layer Transformer. As with Transformers, we add a readout layer that predicts the $\hat{y}_{t+1}^{\mathrm{LSTM}}$ from the final hidden state at the final layer, $\boldsymbol{H}_{:,2t+1}^{(L)}$.

### 3.4 Measuring Algorithmic Similarity

We propose two metrics to measure the similarity between linear regression algorithms.

**Similarity of Errors.** For a linear regression algorithm $\mathcal{A}$, let $\mathcal{A}(\boldsymbol{x}_{t+1} \mid \{\boldsymbol{x}_i, y_i\}_{i=1}^t)$ denote its prediction on the $(t+1)$-th in-context example $\boldsymbol{x}_{t+1}$ after observing the first $t$ examples (see Figure 2). We write $\mathcal{A}(\boldsymbol{x}_{t+1}) := \mathcal{A}(\boldsymbol{x}_{t+1} \mid \{\boldsymbol{x}_i, y_i\}_{i=1}^t)$ for brevity. The errors (i.e., residuals) on the data sequence are:[1]

$$
\mathcal{E}(\mathcal{A} \mid \{\boldsymbol{x}_i, y_i\}_{i=1}^{n+1}) = \left[\mathcal{A}(\boldsymbol{x}_2) - y_2, \cdots, \mathcal{A}(\boldsymbol{x}_{n+1}) - y_{n+1}\right]^\top \in \mathbb{R}^n.
\tag{9}
$$

For any two algorithms $\mathcal{A}_a$ and $\mathcal{A}_b$, their similarity of errors, corresponding to the metric $\mathcal{C}(\cdot, \cdot)$, is

$$
\mathrm{SimE}(\mathcal{A}_a, \mathcal{A}_b) = \mathop{\mathbb{E}}_{\{\boldsymbol{x}_i, y_i\}_{i=1}^{n+1} \sim P_{\mathcal{D}}} \mathcal{C}\Big(\mathcal{E}(\mathcal{A}_a \mid \{\boldsymbol{x}_i, y_i\}_{i=1}^{n+1}), \mathcal{E}(\mathcal{A}_b \mid \{\boldsymbol{x}_i, y_i\}_{i=1}^{n+1})\Big)
\tag{10}
$$

where we use the cosine similarity as our correlation metric $\mathcal{C}(\boldsymbol{u}, \boldsymbol{v}) = \frac{\langle \boldsymbol{u}, \boldsymbol{v} \rangle}{\|\boldsymbol{u}\|_2 \|\boldsymbol{v}\|_2}$. Here $n$ is the total number of in-context examples and $P_{\mathcal{D}}$ is the data generation process discussed previously.

**Similarity of Induced Weights.** All standard algorithms for linear regression estimate a weight vector $\hat{\boldsymbol{w}}$. While neural ICL models like Transformers do not explicitly learn such a weight vector, similar to Akyürek et al. (2022), we can *induce* an implicit weight vector $\tilde{\boldsymbol{w}}$ learned by any algorithm $\mathcal{A}$ by fitting a weight vector to its predictions. To do this, for any fixed sequence of $t$ in-context examples $\{\boldsymbol{x}_i, y_i\}_{i=1}^t$, we sample $T \gg d$ query examples $\tilde{\boldsymbol{x}}_k \overset{\text{i.i.d.}}{\sim} \mathcal{N}(\boldsymbol{0}, \boldsymbol{\Sigma})$, where $k \in [T]$. For this fixed sequence of in-context examples $\{\boldsymbol{x}_i, y_i\}_{i=1}^t$, we create $T$ in-context prediction tasks and use the algorithm $\mathcal{A}$ to make predictions $\mathcal{A}(\tilde{\boldsymbol{x}}_k \mid \{\boldsymbol{x}_i, y_i\}_{i=1}^t)$. We define the induced data matrix and labels as

$$
\tilde{\boldsymbol{X}} = \begin{bmatrix} \tilde{\boldsymbol{x}}_1^\top \\ \vdots \\ \tilde{\boldsymbol{x}}_T^\top \end{bmatrix} \qquad \tilde{\boldsymbol{Y}} = \begin{bmatrix} \mathcal{A}(\tilde{\boldsymbol{x}}_1 \mid \{\boldsymbol{x}_i, y_i\}_{i=1}^t) \\ \vdots \\ \mathcal{A}(\tilde{\boldsymbol{x}}_T \mid \{\boldsymbol{x}_i, y_i\}_{i=1}^t) \end{bmatrix}.
\tag{11}
$$

Then, the induced weight vector for $\mathcal{A}$ and these $t$ in-context examples is:

$$
\tilde{\boldsymbol{w}}_t(\mathcal{A}) := \tilde{\boldsymbol{w}}_t(\mathcal{A} \mid \{\boldsymbol{x}_i, y_i\}_{i=1}^t) = (\tilde{\boldsymbol{X}}^\top \tilde{\boldsymbol{X}})^{-1} \tilde{\boldsymbol{X}}^\top \tilde{\boldsymbol{Y}}.
\tag{12}
$$

---

[1] the indices start from 2 to $n+1$ because we evaluate all cases where $t$ can choose from $1, \cdots, n$.

We measure the similarity between two algorithms $\mathcal{A}_a$ and $\mathcal{A}_b$ by measuring the similarity of induced weight vectors $\tilde{\boldsymbol{w}}_t(\mathcal{A}_a)$ and $\tilde{\boldsymbol{w}}_t(\mathcal{A}_b)$. We define the similarity of induced weights between two algorithms as

$$\text{SimW}(\mathcal{A}_a, \mathcal{A}_b) = \mathop{\mathbb{E}}_{\{\boldsymbol{x}_i, y_i\}_{i=1}^n \sim P_{\mathcal{D}}} \frac{1}{n} \sum_{t=1}^n \mathcal{C}\Big(\tilde{\boldsymbol{w}}_t(\mathcal{A}_a \mid \{\boldsymbol{x}_i, y_i\}_{i=1}^t), \tilde{\boldsymbol{w}}_t(\mathcal{A}_b \mid \{\boldsymbol{x}_i, y_i\}_{i=1}^t))\Big).$$

Each algorithm we consider has its own hyper-parameter(s), for example the number of iterations for Iterative Newton and Gradient Descent, and the number of layers for Transformers (see Section 4.1). When comparing two algorithms, given a choice of hyper-parameters for the first algorithm, we compare with the hyper-parameters for the second algorithm that maximize algorithmic similarity. In other words, we measure whether there exists hyperparameters for the second algorithm that make the two algorithms are similar.

**Definition 4** (Hyper-Parameter Matching). *Let $\mathcal{M}$ be the metric for evaluating similarities between two algorithms $\mathcal{A}_a$ and $\mathcal{A}_b$, which have hyper-parameters $p_a \in \mathcal{P}_a$ and $p_b \in \mathcal{P}_b$, respectively. For a given choice of $p_a$, We define the best-matching hyper-parameters of algorithm $\mathcal{A}_b$ for $\mathcal{A}_a$ as:*

$$p_b^{\mathcal{M}}(p_a) := \underset{p_b \in \mathcal{P}_b}{\arg\min}\, \mathcal{M}(\mathcal{A}_a(\cdot \mid p_a), \mathcal{A}_b(\cdot \mid p_b)). \tag{13}$$

The matching processes can be visualized as heatmaps as shown in Figure 3, where best-matching hyperparameters are highlighted. This enables us to better compare the rate of convergence of algorithms and we will discuss these results further in §4.

## 4 EXPERIMENTAL EVIDENCE

We mainly work on the Transformers-based GPT-2 model with 12 layers with 8 heads per layer. Alternative configurations with fewer heads per layer also support our findings and we defer them to Appendix A. We initially focus on isotropic cases where $\boldsymbol{\Sigma} = \boldsymbol{I}$ and later consider ill-conditioned $\boldsymbol{\Sigma}$ in §4.3. Our training setup is exactly the same as Garg et al. (2022): models are trained with at most $n = 40$ in-context examples for $d = 20$ (with the same learning rate, batch size etc.). We claim that Transformers learn high-order optimization methods in-context.

### 4.1 TRANSFORMERS IMPROVE PROGRESSIVELY OVER LAYERS

Many known algorithms for linear regression, including GD, OGD, and Iterative Newton, are *iterative*: their performance progressively improves as they perform more iterations, eventually converging to a final solution. How could a Transformer implement such an iterative algorithm? von Oswald et al. (2022) propose that deeper *layers* of the Transformer may correspond to more iterations of an iterative method; in particular, they show that there exist Transformer parameters such that each attention layer performs one step of Gradient Descent.

Following this intuition, we first investigate whether the predictions of a trained Transformer improve as the layer index $\ell$ increases. For each layer of hidden states $\boldsymbol{H}^{(\ell)}$ (defined in definition 2), we re-train the `ReadOut` layer to predict $y_t$ for each $t$; the new predictions are given by $\text{ReadOut}^{(\ell)}\big[\boldsymbol{H}^{(\ell)}\big]$. Thus for each input prompt, there are $L$ Transformer predictions parameterized by layer index $\ell$. All parameters besides the `Readout` layer parameters are kept frozen.

As shown in Figure 1a, as we increase the layer index $\ell$, the prediction performance improves progressively. Hence, Transformers progressively improve their predictions over layers $\ell$, similar to how iterative algorithms improve over steps.

### 4.2 TRANSFORMERS ARE MORE SIMILAR TO ITERATIVE NEWTON'S METHOD

Next, we test the more specific hypothesis that the iterative updates performed across Transformer layers are similar to the iterative updates for known iterative algorithms. Specifically, we test whether each layer $\ell$ of the Transformer corresponds to performing $k$ steps of some iterative algorithm, for some $k$ depending on $\ell$. We focus here on Gradient Descent and Iterative Newton's Method; we will discuss online algorithms in Section 4.4.

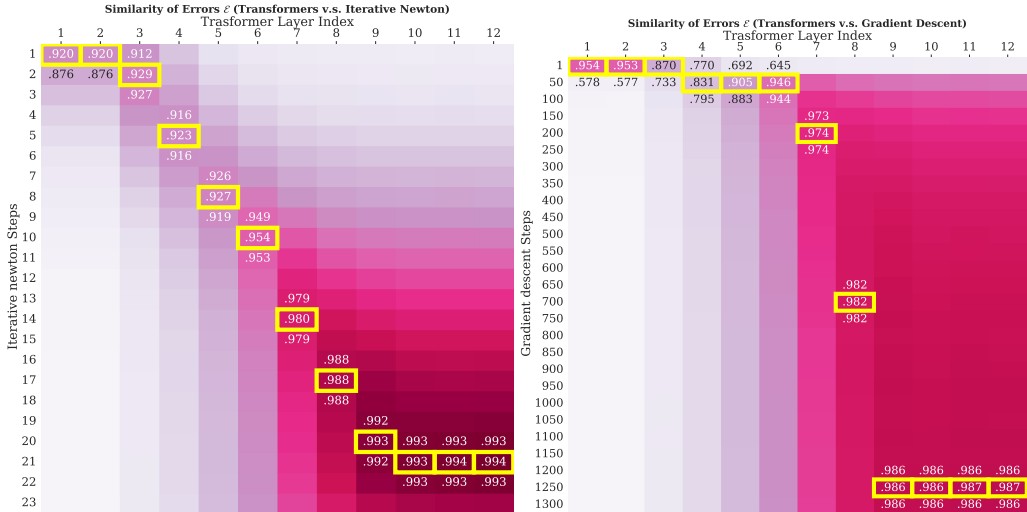

Figure 3: **Heatmaps of Similarity.** The best matching hyper-parameters are highlighted in yellow. See Figure 8 for an additional heatmap where Gradient Descent's steps are shown in log scale.

For each layer $\ell$ of the Transformer, we measure the best-matching similarity (see Def. 4) with candidate iterative algorithms with the optimal choice of the number of steps $k$. As shown in Figure 3, the Transformer has very high error similarity with Iterative Newton's method at all layers. Moreover, we see a clear *linear* trend between layer 3 and layer 9 of the Transformer, where each layer appears to compute roughly 3 additional iterations of the Iterative Newton's method. This trend only stops at the last few layers because both algorithms converge to the OLS solution; Newton is known to converge to OLS (see §3.2), and we verify in Appendix A.1 that the last few layers of the Transformer also basically compute OLS (see Figure 10 in the Appendix). We observe the same trends when using similarity of induced weights as our similarity metric (see Figure 7 in the Appendix),

In contrast, even though GD has a comparable similarity with the Transformers at later layers, their best matching follows an *exponential* trend. For well-conditioned problems, to achieve $\epsilon$ error, the convergence rate of GD is $\mathcal{O}(\log(1/\epsilon))$ while the convergence rate of Newton is $\mathcal{O}(\log\log(1/\epsilon))$ (see §3.2). Therefore the convergence rate of Newton is exponentially faster than GD. Transformer's *linear* correspondence with Newton and its *exponential* correspondence with GD provides strong evidence that the rate of convergence of Transformers is similar to Newton, i.e., $\mathcal{O}(\log\log(1/\epsilon))$.

Overall, we conclude that a Transformer trained to perform in-context linear regression learns to implement an algorithm that is very similar to Iterative Newton's method, not Gradient Descent. Starting at layer 3, subsequent layers of the Transformer compute more and more iterations of Iterative Newton's method. This algorithm successfully solves the linear regression problem, as it converges to the optimal OLS solution in the final layers.

### 4.3 TRANSFORMERS PERFORM WELL ON ILL-CONDITIONED DATA

We repeat the same experiments with data $x_i \overset{\text{i.i.d.}}{\sim} \mathcal{N}(\mathbf{0}, \mathbf{\Sigma})$ sampled from an ill-condition covariance matrix $\mathbf{\Sigma}$ with condition number $\kappa(\mathbf{\Sigma}) = 100$, and eigenbasis chosen uniformly at random. The first $d/2$ eigenvalues of $\mathbf{\Sigma}$ are 100, the last $d/2$ are 1.

As shown in Figure 4, the Transformer model's performance still closely matches Iterative Newton's Method with 21 iterations, same as when $\mathbf{\Sigma} = \mathbf{I}$ (see layer 10-12 in Figure 3). The convergence of higher-order methods has a mild logarithmic dependence on the condition number since they correct for the curvature. On the other hand, Gradient Descent's convergence is affected polynomially by conditioning. As $\kappa(\mathbf{\Sigma})$ increase from 1 to 100, the number steps required for GD's convergence increases

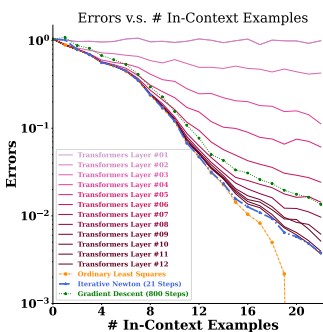

Figure 4: Transformers performance on ill-conditioned data.

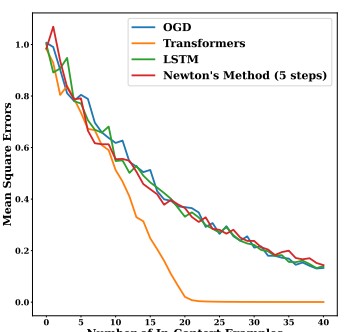 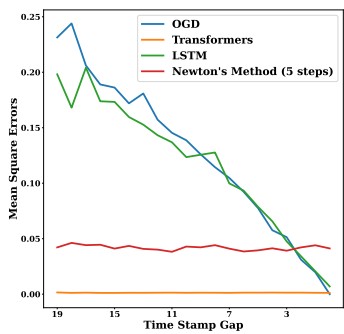

Table 1: **Similarity of errors between algorithms.** Transformers are more similar to full-observation methods such as Newton and GD; and LSTMs are more similar to online methods such as OGD.

|         | Transformers | LSTM  |
| ------- | ------------ | ----- |
| Newton  | **0.991**    | 0.920 |
| GD      | **0.957**    | 0.916 |
| OGD     | 0.806        | **0.954** |

Figure 5: In the left figure, we measure model predictions with normalized MSE. Though LSTM is seemingly most similar to Newton's Method with only 5 steps, neither algorithm converges yet. OGD also has a similar trend as LSTM. In the center figure, we measure model's forgetting phenomenon (see Appendix A.4 for explanations), and find both Transformers and not-converged Newton have better memorization than LSTM and OGD. In the right table, we find that Transformers are more similar to Newton and GD than LSTMs do while LSTM is significantly more similar to OGD.

significantly (see Fig. 4 where GD requires 800 steps to converge), making it impossible for a 12-layer Transformers to implement these many gradient updates. We also note that preconditioning the data by $(\boldsymbol{X}^{\top}\boldsymbol{X})^{\dagger}$ can make the data well-conditioned, but since the eigenbasis is chosen uniformly at random, with high probability there is no sparse pre-conditioner or any fixed pre-conditioner which works across the data distribution. Computing $(\boldsymbol{X}^{\top}\boldsymbol{X})^{\dagger}$ appears to be as hard as computing the OLS solution (Eq. 4)—in fact Sharan et al. (2019) conjecture that first-order methods such as gradient descent and its variants cannot avoid polynomial dependencies in condition number in the ill-conditioned case.

See Appendix A.2 for detailed experiments on ill-conditioned problems. These experiments on ill-conditioned data further strengthen our hypothesis that Transformers are learning to perform higher-order optimization methods in-context, not Gradient Descent.

## 4.4 LSTM IS MORE SIMILAR TO OGD THAN TRANSFORMERS

As discussed in Section 3, LSTM is an alternative auto-regressive model widely used before the introduction of Transformers. Thus, a natural research question is: *If Transformers can learn in-context, can LSTMs do so as well? If so, do they learn the same algorithms?* To answer this question, we train a 10-layer LSTM model, with 5.3M parameters, in an identical manner to the Transformers (with 9.5M parameters) studied in the previous sections.[2]

Figure 5 plots the mean squared error of Transformers, LSTMs, and other standard methods as a function of the number of in-context (i.e., training) examples provided. While LSTMs can also learn linear regression in-context, they have much higher mean-squared error than Transformers. Their error rate is similar to Iterative Newton's Method after only 5 iterations, a point where it is far from converging to the OLS solution.

LSTMs' inferior performance to Transformers can be explained by the inability of LSTMs to use deeper layers to improve their predictions. Figure 1 shows that LSTM performance does not improve across layers—a readout head fine-tuned for the first layer makes equally good predictions as the full 10-layer model. Thus, LSTMs seem poorly equipped to fully implement iterative algorithms.

Finally, we show that LSTMs behave more like an online learning algorithm than Transformers. In particular, its predictions are biased towards getting more recent training examples correct, as opposed to earlier examples, as shown in Figure 5. This property makes LSTMs similar to online Gradient Descent. In contrast, five steps of Newton's method has the same error on average for recent and early examples, showing that the LSTM implements a very different algorithm from a few iterations of Newton. Similarly, Table 1 shows that LSTMs are more similar to OGD than

---

[2]While the LSTM has fewer parameters than the Transformer, we found in preliminary experiments that increasing the hidden dimension or number of layers in the LSTM would not substantively change our results.

Transformers are, whereas Transformers are more similar to Newton and GD than LSTMs. We hypothesize that since LSTMs have limited memory, they must learn in a roughly online fashion; in contrast, Transformers' attention heads can access the entire sequence of past examples, enabling it to learn more complex algorithms.

## 5 MECHANISTIC EVIDENCE

Our empirical evidence demonstrates that Transformers behave much more similarly to Iterative Newton's than to Gradient Descent. Iterative Newton is a higher-order optimization method, and is algorithmically more involved than Gradient Descent. We begin by first examining this difference in complexity. As discussed in Section 3, the updates for Iterative Newton are of the form,

$$\hat{\boldsymbol{w}}_{k+1}^{\text{Newton}} = \boldsymbol{M}_{k+1}\boldsymbol{X}^\top\boldsymbol{y} \quad \text{where} \quad \boldsymbol{M}_{k+1} = 2\boldsymbol{M}_k - \boldsymbol{M}_k\boldsymbol{S}\boldsymbol{M}_k \tag{14}$$

and $\boldsymbol{M}_0 = \alpha\boldsymbol{S}$ for some $\alpha > 0$. We can express $\boldsymbol{M}_k$ in terms of powers of $\boldsymbol{S}$ by expanding iteratively, for example $\boldsymbol{M}_1 = 2\alpha\boldsymbol{S} - 4\alpha^2\boldsymbol{S}^3, \boldsymbol{M}_2 = 4\alpha\boldsymbol{S} - 12\alpha^2\boldsymbol{S}^3 + 16\alpha^3\boldsymbol{S}^5 - 16\alpha^4\boldsymbol{S}^7$, and in general $\boldsymbol{M}_k = \sum_{s=1}^{2^{k+1}-1}\beta_s\boldsymbol{S}^s$ for some $\beta_s \in \mathbb{R}$ (see Appendix B.3 for detailed calculations). Note that $k$ steps of Iterative Newton's requires computing $\Omega(2^k)$ moments of $\boldsymbol{S}$. Let us contrast this with Gradient Descent. Gradient Descent updates for linear regression take the form,

$$\hat{\boldsymbol{w}}_{k+1}^{\text{GD}} = \hat{\boldsymbol{w}}_k^{\text{GD}} - \eta(\boldsymbol{S}\hat{\boldsymbol{w}}_k^{\text{GD}} - \boldsymbol{X}^\top\boldsymbol{y}). \tag{15}$$

Like Iterative Newton, we can express $\hat{\boldsymbol{w}}_k^{\text{GD}}$ in terms of powers of $\boldsymbol{S}$ and $\boldsymbol{X}^\top\boldsymbol{y}$. However, after $k$ steps of Gradient Descent, the highest power of $\boldsymbol{S}$ is only $O(k)$. This exponential separation is consistent with the exponential gap in terms of the parameter dependence in the convergence rate—$\mathcal{O}(\kappa(\boldsymbol{S})\log(1/\epsilon))$ steps for Gradient Descent compared to $\mathcal{O}(\log\kappa(\boldsymbol{S}) + \log\log(1/\epsilon))$ steps for Iterative Newton. Therefore, a natural question is whether Transformers can actually represent as complicated of a method as Iterative Newton with only polynomially many layers?

Theorem 1 shows that this is indeed possible.

**Theorem 1.** *There exist Transformer weights such that on any set of in-context examples $\{\boldsymbol{x}_i, y_i\}_{i=1}^n$ and test point $\boldsymbol{x}_{\text{test}}$, the Transformer predicts on $\boldsymbol{x}_{\text{test}}$ using $\boldsymbol{x}_{\text{test}}^\top\hat{\boldsymbol{w}}_k^{\text{Newton}}$. Here $\hat{\boldsymbol{w}}_k^{\text{Newton}}$ are the Iterative Newton updates given by $\hat{\boldsymbol{w}}_k^{\text{Newton}} = \boldsymbol{M}_k\boldsymbol{X}^\top\boldsymbol{y}$ where $\boldsymbol{M}_j$ is updated as*

$$\boldsymbol{M}_j = 2\boldsymbol{M}_{j-1} - \boldsymbol{M}_{j-1}\boldsymbol{S}\boldsymbol{M}_{j-1}, 1 \le j \le k, \quad \boldsymbol{M}_0 = \alpha\boldsymbol{S},$$

*for some $\alpha > 0$ and $\boldsymbol{S} = \boldsymbol{X}^\top\boldsymbol{X}$. The number of layers of the transformer is $\mathcal{O}(k)$ and the dimensionality of the hidden layers is $\mathcal{O}(d)$.*

We note that our proof uses full attention instead of causal attention and ReLU activations for the self-attention layers. The definitions of these and the full proof appear in Appendix B.

## 6 CONCLUSION AND DISCUSSION

In this work, we studied how Transformers perform in-context learning for linear regression. In contrast with the hypothesis that Transformers learn in-context by implementing gradient descent, our experimental results show that different Transformer layers match iterations of Iterative Newton's method *linearly* and Gradient Descent *exponentially*. This suggests that Transformers share a similar rate of convergence to Iterative Newton's method but not Gradient Descent. Moreover, Transformers can perform well empirically on ill-conditioned linear regression, whereas first-order methods such as Gradient Descent struggle. Theoretically, we provide exact Transformer circuits that can implement $k$-steps of Iterative Newton's method with $\mathcal{O}(k)$ layers to make predictions from in-context examples.

An interesting direction is to explore a wider range of higher-order methods that Transformers can implement. It also seems promising to extend our analysis to classification problems, especially given recent work showing that Transformers resemble SVMs in classification tasks (Li et al., 2023; Tarzanagh et al., 2023b;a). Finally, a natural question is to understand the differences in the model architecture that make Transformers better in-context learners than LSTMs. Based on our investigations with LSTMs, we hypothesize that Transformers can implement more powerful algorithms because of having access to a longer history of examples. Investigating the role of this additional memory in learning appears to be an intriguing direction.

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

# APPENDIX

## A ADDITIONAL EXPERIMENTAL RESULTS

### A.1 ADDITIONAL RESULTS ON ISOTROPIC DATA

#### A.1.1 HEATMAPS

We present heatmaps with all values of similarities.

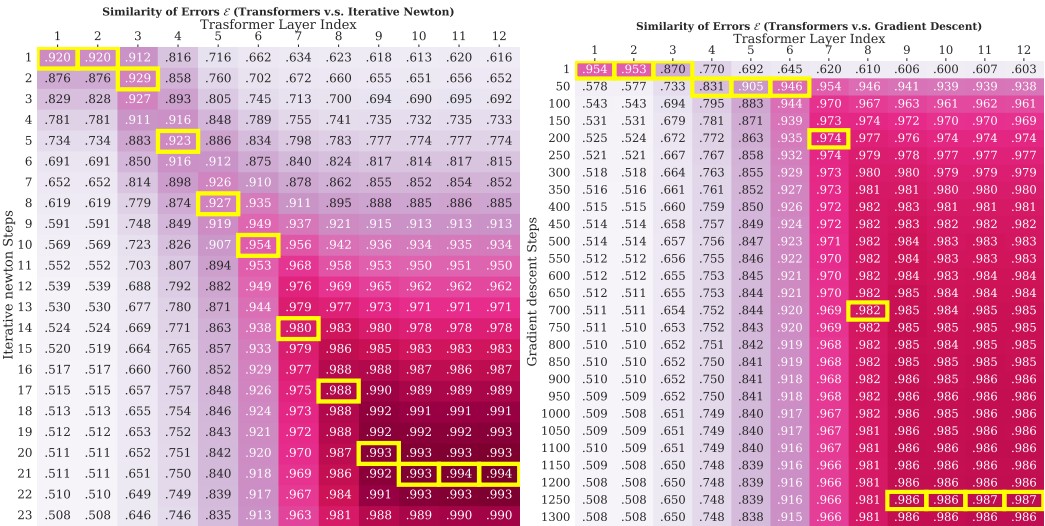

Figure 6: **Similarity of Errors.** The best matching hyper-parameters are highlighted in yellow.

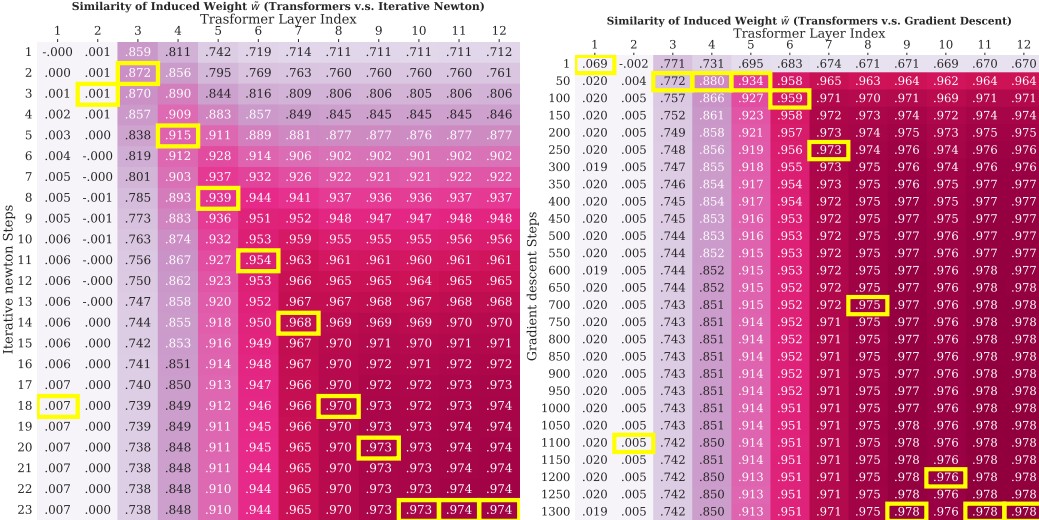

Figure 7: **Similarity of Induced Weight Vectors.** The best matching hyper-parameters are highlighted in yellow.

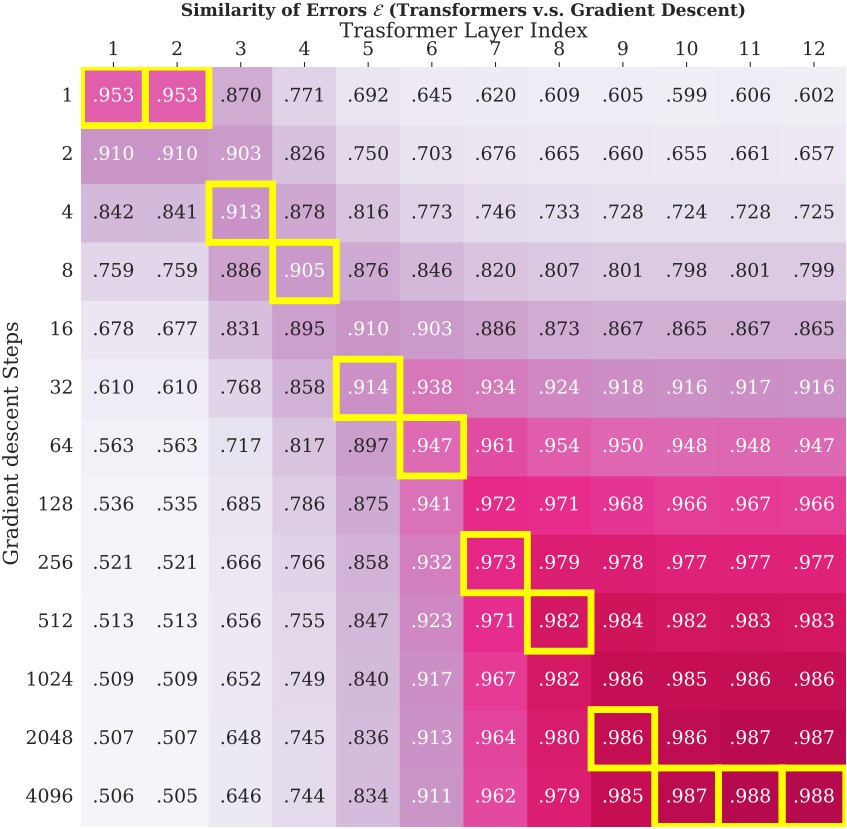

Figure 8: **Similarity of Errors of Gradient Descent in Log Scale.** The best matching hyperparameters are highlighted in yellow. Putting the number of steps of Gradient Descent in log scale further verifies the claim that Transformer's rate of covergence is exponentially faster than that of Gradient Descent.

## A.1.2 ADDITIONAL RESULTS ON COMPARISON OVER TRANSFORMER LAYERS

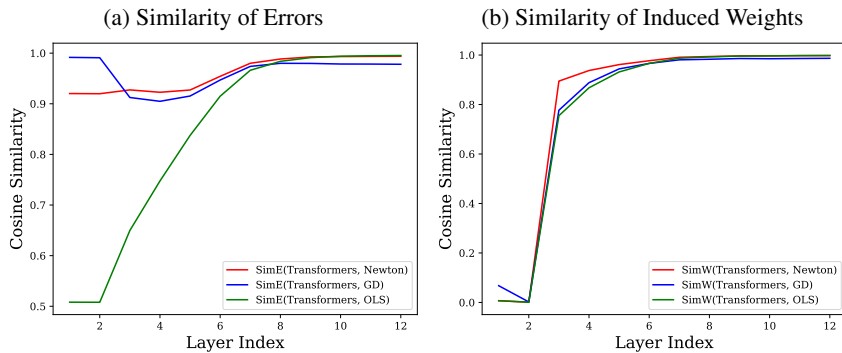

Figure 9: Similarities between Transformer and candidate algorithms. Transformers resemble *Iterative Newton's Method* the most.

### A.1.3 ADDITIONAL RESULTS ON SIMILARITY OF INDUCED WEIGHTS

We present more details line plots for how the similarity of weights changes as the models see more in-context observations $\{\boldsymbol{x}_i, y_i\}_{i=1}^n$, i.e., as $n$ increases. We fix the number of Transformers layers $\ell$ and compare with other algorithms with their best-match hyperparamters to $\ell$ in Figure 10.

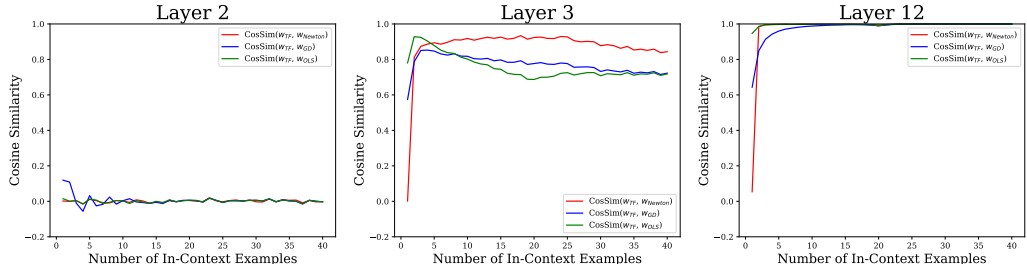

Figure 10: *Similarity of induced weights* over varying number of in-context examples, on three layer indices of Transformers, indexed as 2, 3 and 12. We find that initially at layer 2, the Transformers model hasn't learned so it has zero similarity to all candidate algorithms. As we progress to the next layer number 3, we find that Transformers start to learn, and when provided few examples, Transformers are more similar to OLS but soon become most similar to the Iterative Newton's Method. Layer 12 shows that Transformers in the later layers converge to the OLS solution when provided more than 1 example. We also find there is a dip around $n = d$ for similarity between Transformers and OLS but not for Transformers and Newton, and this is probably because OLS has a more prominent double-descent phenomenon than Transformers and Newton.

## A.2 Experiments on Ill-Conditioned Problems

In this section, we repeat the same experiments as we did on isotropic data in the main text and in Appendix A.1, and we change the covariance matrix to be ill-conditioned such that $\kappa(\Sigma) = 100$.

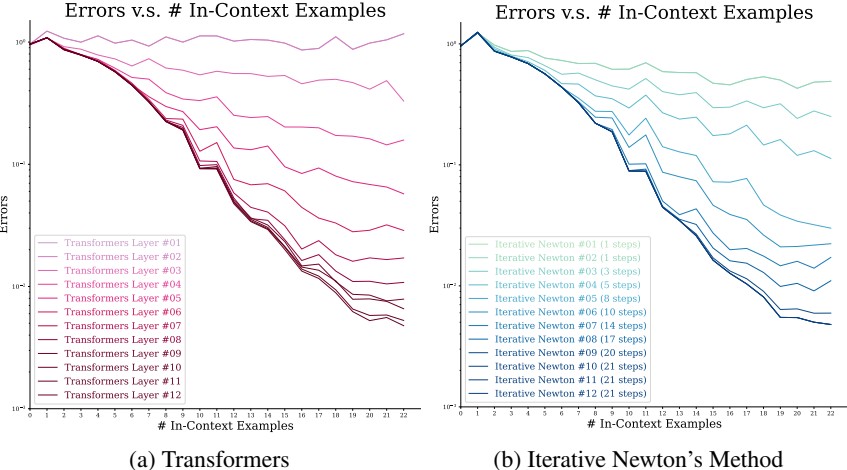

(a) Transformers        (b) Iterative Newton's Method

Figure 11: **Progression of Algorithms on Ill-Conditioned Data.** Transformer's performance still improves over the layer index $\ell$; Iterative Newton's Method's performance improves over the number of iterations $t$ and we plot the best-matching $t$ to Transformer's $\ell$ following definition 4.

We also present the heatmaps to find the best-matching hyper-parameters and conclude that Transformers are similar to Newton's method than GD in ill-conditioned data.

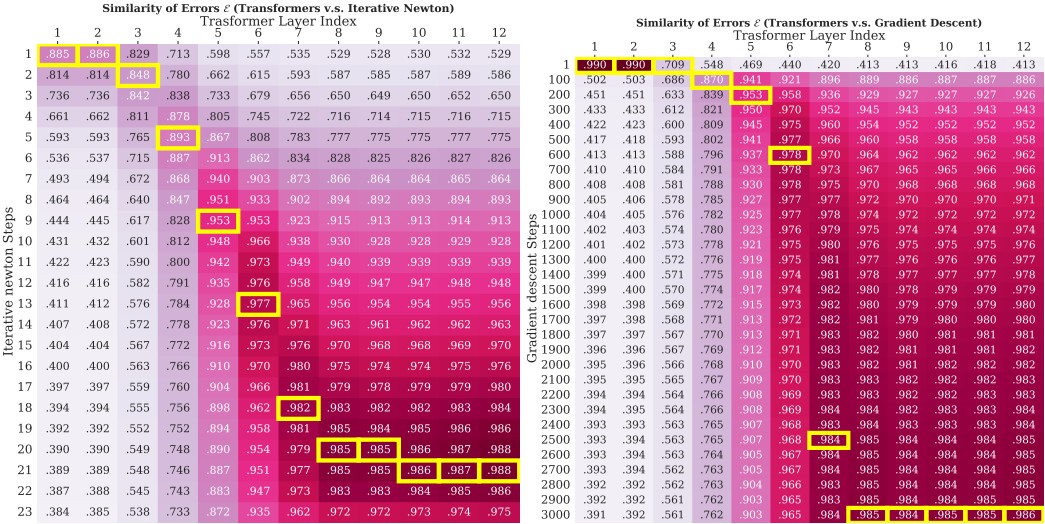

Figure 12: **Similarity of Errors on Ill-Conditioned Data.** The best matching hyper-parameters are highlighted in yellow.

**Similarity of Induced Weight $\hat{w}$ (Transformers v.s. Iterative Newton)**

Trasformer Layer Index

| Iterative newton Steps | 1 | 2 | 3 | 4 | 5 | 6 | 7 | 8 | 9 | 10 | 11 | 12 |
|---|---|---|---|---|---|---|---|---|---|---|---|---|
| 1 | .003 | .023 | .646 | .739 | .747 | .721 | .650 | .626 | .617 | .615 | .612 | .608 |
| 2 | .003 | .024 | .659 | .778 | .793 | .765 | .690 | .664 | .654 | .653 | .649 | .645 |
| 3 | .003 | .024 | .662 | .808 | .834 | .805 | .726 | .699 | .688 | .687 | .683 | .679 |
| 4 | .002 | .024 | .655 | .827 | .868 | .838 | .755 | .728 | .717 | .715 | .711 | .707 |
| 5 | .002 | .024 | .644 | .836 | .893 | .864 | .779 | .751 | .740 | .738 | .734 | .729 |
| 6 | .002 | .024 | .632 | .838 | .907 | .881 | .795 | .766 | .755 | .753 | .749 | .744 |
| 7 | .002 | .023 | .622 | .835 | .915 | .892 | .805 | .777 | .765 | .764 | .760 | .755 |
| 8 | .001 | .023 | .615 | .831 | .919 | .900 | .814 | .785 | .773 | .772 | .768 | .763 |
| 9 | .002 | .023 | .610 | .827 | .920 | .906 | .820 | .792 | .780 | .779 | .775 | .770 |
| 10 | .002 | .023 | .606 | .824 | .920 | .911 | .828 | .800 | .788 | .787 | .783 | .778 |
| 11 | .002 | .023 | .603 | .821 | .919 | .917 | .837 | .810 | .798 | .797 | .793 | .789 |
| 12 | .002 | .023 | .599 | .816 | .917 | .924 | .851 | .826 | .813 | .812 | .809 | .804 |
| 13 | .002 | .023 | .592 | .807 | .910 | .930 | .868 | .846 | .834 | .832 | .829 | .825 |
| 14 | .002 | .022 | .579 | .791 | .896 | .932 | .885 | .868 | .856 | .856 | .853 | .849 |
| 15 | .002 | .021 | .562 | .768 | .873 | .926 | .897 | .889 | .878 | .877 | .876 | .873 |
| 16 | .002 | .021 | .544 | .744 | .849 | .914 | .904 | .906 | .895 | .894 | .895 | .893 |
| 17 | .002 | .020 | .528 | .722 | .826 | .900 | .905 | .918 | .909 | .908 | .910 | .909 |
| 18 | .002 | .020 | .515 | .704 | .807 | .886 | .902 | .925 | .918 | .918 | .922 | .921 |
| 19 | .003 | .019 | .505 | .690 | .792 | .873 | .897 | .929 | .924 | .925 | .929 | .929 |
| 20 | .003 | .019 | .498 | .680 | .781 | .863 | .891 | .931 | .926 | .928 | .933 | .933 |
| 21 | .003 | .019 | .492 | .672 | .772 | .854 | .885 | .930 | .927 | .929 | .935 | .935 |
| 22 | .003 | .019 | .488 | .666 | .766 | .848 | .880 | .929 | .926 | .929 | .935 | .935 |
| 23 | .003 | .019 | .485 | .662 | .761 | .843 | .877 | .927 | .925 | .927 | .934 | .935 |

**Similarity of Induced Weight $\hat{w}$ (Transformers v.s. Gradient Descent)**

Trasformer Layer Index

| Gradient descent Steps | 1 | 2 | 3 | 4 | 5 | 6 | 7 | 8 | 9 | 10 | 11 | 12 |
|---|---|---|---|---|---|---|---|---|---|---|---|---|
| 1 | .010 | -.062 | .292 | .337 | .346 | .333 | .294 | .287 | .280 | .284 | .273 | .274 |
| 100 | .009 | .010 | .625 | .829 | .913 | .907 | .831 | .807 | .798 | .796 | .790 | .787 |
| 200 | .008 | .011 | .611 | .821 | .916 | .924 | .858 | .839 | .830 | .828 | .822 | .820 |
| 300 | .008 | .012 | .602 | .812 | .912 | .931 | .874 | .858 | .850 | .848 | .843 | .841 |
| 400 | .008 | .012 | .595 | .804 | .906 | .934 | .884 | .873 | .864 | .862 | .858 | .857 |
| 500 | .008 | .012 | .589 | .796 | .899 | .935 | .892 | .883 | .875 | .873 | .870 | .868 |
| 600 | .008 | .013 | .583 | .789 | .893 | .934 | .897 | .892 | .884 | .882 | .879 | .878 |
| 700 | .008 | .013 | .577 | .782 | .886 | .932 | .901 | .898 | .891 | .889 | .886 | .885 |
| 800 | .008 | .013 | .572 | .775 | .880 | .930 | .904 | .904 | .896 | .895 | .892 | .892 |
| 900 | .008 | .014 | .568 | .769 | .875 | .928 | .906 | .908 | .901 | .899 | .898 | .897 |
| 1000 | .008 | .014 | .564 | .764 | .870 | .926 | .908 | .912 | .905 | .903 | .902 | .901 |
| 1100 | .007 | .014 | .560 | .759 | .865 | .924 | .909 | .915 | .908 | .907 | .906 | .905 |
| 1200 | .007 | .014 | .557 | .755 | .860 | .922 | .910 | .918 | .911 | .910 | .909 | .909 |
| 1300 | .007 | .014 | .554 | .750 | .856 | .919 | .910 | .920 | .914 | .912 | .912 | .911 |
| 1400 | .007 | .014 | .551 | .747 | .852 | .917 | .911 | .922 | .916 | .914 | .914 | .914 |
| 1500 | .007 | .014 | .548 | .743 | .848 | .915 | .911 | .923 | .918 | .916 | .916 | .916 |
| 1600 | .007 | .014 | .546 | .740 | .845 | .913 | .911 | .925 | .919 | .918 | .918 | .918 |
| 1700 | .007 | .015 | .544 | .737 | .842 | .911 | .911 | .926 | .921 | .920 | .920 | .920 |
| 1800 | .007 | .015 | .542 | .734 | .839 | .909 | .911 | .927 | .922 | .921 | .921 | .922 |
| 1900 | .008 | .015 | .540 | .731 | .836 | .907 | .910 | .928 | .923 | .922 | .923 | .923 |
| 2000 | .007 | .015 | .538 | .729 | .834 | .906 | .910 | .929 | .924 | .923 | .924 | .924 |
| 2100 | .007 | .015 | .536 | .726 | .831 | .904 | .910 | .930 | .925 | .924 | .925 | .926 |
| 2200 | .007 | .015 | .534 | .724 | .829 | .902 | .910 | .931 | .926 | .925 | .926 | .927 |
| 2300 | .007 | .015 | .533 | .722 | .827 | .901 | .909 | .931 | .927 | .927 | .927 | .928 |
| 2400 | .007 | .015 | .531 | .720 | .825 | .900 | .909 | .932 | .928 | .927 | .928 | .929 |
| 2500 | .007 | .015 | .530 | .718 | .822 | .898 | .909 | .932 | .928 | .927 | .929 | .929 |
| 2600 | .007 | .015 | .529 | .716 | .821 | .897 | .908 | .933 | .929 | .928 | .929 | .930 |
| 2700 | .007 | .015 | .528 | .715 | .819 | .896 | .908 | .933 | .929 | .929 | .930 | .931 |
| 2800 | .007 | .015 | .527 | .713 | .817 | .894 | .908 | .934 | .930 | .929 | .931 | .932 |
| 2900 | .007 | .015 | .525 | .712 | .816 | .893 | .907 | .934 | .930 | .929 | .931 | .932 |
| 3000 | .007 | .015 | .524 | .710 | .814 | .892 | .907 | .934 | .931 | .930 | .932 | .933 |

Figure 13: **Similarity of Induced Weights on Ill-Conditioned Data.** The best matching hyper-parameters are highlighted in yellow.

## A.3 EXPERIMENTS ON ALTERNATIVE CONFIGURATIONS

In this section, we present experimental results from an alternative model configurations than the main text. We show in the main text that Transformers learn higher-order optimization methods in-context where the experiments are using a GPT-2 model with 12 layers and 8 heads per layer. In this section, we present experiments with a GPT-2 model with 12 layers but only 1 head per layer.

**Similarity of Errors $\mathcal{E}$ (Transformers v.s. Iterative Newton)**

Trasformer Layer Index

| Iterative newton Steps | 1 | 2 | 3 | 4 | 5 | 6 | 7 | 8 | 9 | 10 | 11 | 12 |
|---|---|---|---|---|---|---|---|---|---|---|---|---|
| 1 | .920 | .920 | .911 | .909 | .861 | .785 | .707 | .671 | .647 | .631 | .626 | .619 |
| 2 | .876 | .876 | .892 | .912 | .879 | .823 | .749 | .709 | .685 | .667 | .663 | .655 |
| 3 | .829 | .829 | .864 | .901 | .887 | .859 | .791 | .750 | .726 | .706 | .702 | .694 |
| 4 | .780 | .780 | .829 | .877 | .884 | .887 | .832 | .792 | .768 | .746 | .743 | .735 |
| 5 | .733 | .733 | .791 | .845 | .872 | .906 | .867 | .832 | .810 | .787 | .784 | .776 |
| 6 | .690 | .690 | .753 | .811 | .853 | .913 | .896 | .869 | .849 | .825 | .823 | .816 |
| 7 | .654 | .654 | .719 | .777 | .829 | .910 | .916 | .900 | .884 | .861 | .860 | .852 |
| 8 | .624 | .624 | .688 | .746 | .805 | .900 | .927 | .924 | .912 | .894 | .893 | .885 |
| 9 | .598 | .598 | .661 | .719 | .780 | .885 | .930 | .939 | .934 | .920 | .920 | .913 |
| 10 | .576 | .576 | .637 | .695 | .757 | .867 | .926 | .947 | .947 | .941 | .942 | .935 |
| 11 | .559 | .559 | .619 | .676 | .738 | .851 | .918 | .948 | .955 | .956 | .957 | .951 |
| 12 | .546 | .546 | .605 | .662 | .723 | .837 | .910 | .947 | .958 | .966 | .968 | .963 |
| 13 | .537 | .537 | .595 | .651 | .712 | .826 | .903 | .944 | .959 | .973 | .976 | .972 |
| 14 | .530 | .530 | .587 | .644 | .704 | .817 | .896 | .940 | .958 | .977 | .981 | .979 |
| 15 | .525 | .525 | .582 | .638 | .698 | .810 | .890 | .936 | .957 | .980 | .984 | .984 |
| 16 | .522 | .522 | .578 | .634 | .693 | .806 | .886 | .933 | .955 | .981 | .986 | .987 |
| 17 | .519 | .519 | .576 | .631 | .690 | .802 | .882 | .930 | .953 | .981 | .987 | .989 |
| 18 | .518 | .517 | .573 | .629 | .688 | .799 | .880 | .928 | .951 | .981 | .987 | .991 |
| 19 | .516 | .516 | .572 | .627 | .686 | .798 | .878 | .926 | .949 | .980 | .986 | .992 |
| 20 | .515 | .515 | .571 | .626 | .685 | .796 | .876 | .924 | .948 | .979 | .986 | .992 |
| 21 | .514 | .514 | .570 | .625 | .684 | .795 | .874 | .923 | .946 | .978 | .985 | .992 |
| 22 | .513 | .513 | .569 | .624 | .683 | .793 | .872 | .920 | .945 | .976 | .983 | .991 |
| 23 | .510 | .510 | .565 | .620 | .679 | .787 | .865 | .914 | .938 | .969 | .976 | .984 |

**Similarity of Errors $\mathcal{E}$ (Transformers v.s. Gradient Descent)**

Trasformer Layer Index

| Gradient descent Steps | 1 | 2 | 3 | 4 | 5 | 6 | 7 | 8 | 9 | 10 | 11 | 12 |
|---|---|---|---|---|---|---|---|---|---|---|---|---|
| 1 | .954 | .955 | .915 | .885 | .840 | .757 | .685 | .655 | .630 | .615 | .609 | .604 |
| 50 | .584 | .585 | .645 | .703 | .764 | .870 | .923 | .943 | .945 | .943 | .944 | .939 |
| 100 | .552 | .552 | .610 | .668 | .729 | .841 | .911 | .945 | .955 | .964 | .966 | .962 |
| 150 | .539 | .539 | .596 | .653 | .713 | .826 | .902 | .941 | .956 | .970 | .973 | .971 |
| 200 | .532 | .532 | .588 | .645 | .705 | .819 | .897 | .938 | .955 | .973 | .977 | .975 |
| 250 | .528 | .528 | .583 | .640 | .700 | .813 | .892 | .936 | .954 | .974 | .979 | .978 |
| 300 | .525 | .525 | .581 | .637 | .697 | .810 | .889 | .934 | .953 | .975 | .980 | .979 |
| 350 | .522 | .522 | .578 | .635 | .694 | .807 | .887 | .932 | .952 | .975 | .980 | .980 |
| 400 | .520 | .520 | .576 | .632 | .692 | .804 | .885 | .931 | .951 | .976 | .981 | .982 |
| 450 | .519 | .520 | .575 | .631 | .691 | .803 | .884 | .930 | .950 | .976 | .981 | .983 |
| 500 | .519 | .519 | .574 | .630 | .689 | .801 | .882 | .928 | .950 | .976 | .981 | .983 |
| 550 | .518 | .518 | .573 | .629 | .688 | .801 | .881 | .928 | .949 | .976 | .981 | .983 |
| 600 | .517 | .517 | .572 | .628 | .687 | .799 | .880 | .927 | .948 | .976 | .982 | .984 |
| 650 | .516 | .516 | .572 | .628 | .687 | .799 | .880 | .926 | .948 | .976 | .982 | .985 |
| 700 | .516 | .516 | .571 | .627 | .686 | .798 | .879 | .926 | .947 | .976 | .981 | .985 |
| 750 | .516 | .516 | .570 | .626 | .686 | .797 | .878 | .925 | .947 | .976 | .981 | .985 |
| 800 | .516 | .516 | .571 | .626 | .685 | .797 | .878 | .925 | .947 | .976 | .982 | .985 |
| 850 | .515 | .515 | .570 | .626 | .685 | .796 | .877 | .924 | .947 | .976 | .982 | .985 |
| 900 | .514 | .514 | .569 | .625 | .684 | .795 | .876 | .924 | .946 | .976 | .982 | .985 |
| 950 | .513 | .514 | .568 | .625 | .684 | .795 | .876 | .923 | .946 | .976 | .981 | .986 |
| 1000 | .514 | .514 | .569 | .624 | .683 | .795 | .875 | .923 | .946 | .976 | .982 | .986 |
| 1050 | .513 | .513 | .568 | .624 | .683 | .795 | .875 | .923 | .946 | .976 | .982 | .986 |
| 1100 | .513 | .513 | .568 | .624 | .683 | .794 | .875 | .922 | .945 | .975 | .981 | .986 |
| 1150 | .513 | .513 | .568 | .624 | .683 | .794 | .875 | .922 | .945 | .975 | .981 | .986 |
| 1200 | .513 | .513 | .567 | .623 | .682 | .794 | .874 | .922 | .945 | .975 | .981 | .986 |
| 1250 | .513 | .513 | .567 | .623 | .682 | .794 | .875 | .922 | .945 | .975 | .981 | .986 |
| 1300 | .513 | .513 | .568 | .623 | .682 | .793 | .874 | .921 | .944 | .975 | .981 | .986 |

Figure 14: **Similarity of Errors on an alternative Transformers Configuration.** The best matching hyper-parameters are highlighted in yellow.

**Similarity of Induced Weight $\hat{w}$ (Transformers v.s. Iterative Newton)**

Trasformer Layer Index

| Iterative newton Steps | 1 | 2 | 3 | 4 | 5 | 6 | 7 | 8 | 9 | 10 | 11 | 12 |
|---|---|---|---|---|---|---|---|---|---|---|---|---|
| 1 | .000 | -.003 | .581 | .756 | .740 | .757 | .734 | .717 | .713 | .710 | .711 | .712 |
| 2 | .001 | -.003 | .590 | .767 | .770 | .802 | .782 | .766 | .762 | .759 | .760 | .761 |
| 3 | .002 | -.002 | .588 | .764 | .789 | .840 | .827 | .811 | .807 | .805 | .806 | .807 |
| 4 | .002 | -.002 | .580 | .751 | .797 | .868 | .864 | .850 | .847 | .844 | .845 | .846 |
| 5 | .003 | -.002 | .567 | .734 | .796 | .885 | .891 | .881 | .878 | .876 | .877 | .877 |
| 6 | .003 | -.002 | .553 | .716 | .789 | .893 | .911 | .905 | .903 | .901 | .902 | .902 |
| 7 | .003 | -.001 | .541 | .700 | .780 | .894 | .923 | .922 | .921 | .921 | .922 | .922 |
| 8 | .003 | -.001 | .531 | .686 | .770 | .891 | .930 | .934 | .935 | .936 | .937 | .937 |
| 9 | .003 | -.001 | .522 | .675 | .761 | .886 | .932 | .941 | .944 | .947 | .948 | .948 |
| 10 | .002 | -.000 | .515 | .666 | .753 | .880 | .932 | .945 | .949 | .954 | .955 | .955 |
| 11 | .003 | -.000 | .510 | .660 | .746 | .875 | .930 | .946 | .952 | .959 | .961 | .961 |
| 12 | .003 | .000 | .506 | .655 | .741 | .870 | .928 | .947 | .954 | .963 | .964 | .965 |
| 13 | .003 | .000 | .503 | .652 | .738 | .867 | .926 | .946 | .954 | .965 | .967 | .968 |
| 14 | .003 | .001 | .501 | .650 | .736 | .864 | .924 | .945 | .954 | .966 | .968 | .969 |
| 15 | .003 | .001 | .500 | .648 | .734 | .862 | .922 | .944 | .954 | .967 | .969 | .971 |
| 16 | .003 | .001 | .499 | .647 | .732 | .861 | .921 | .943 | .953 | .968 | .970 | .972 |
| 17 | .003 | .001 | .498 | .646 | .731 | .860 | .920 | .942 | .953 | .968 | .970 | .972 |
| 18 | .003 | .001 | .498 | .645 | .731 | .859 | .919 | .942 | .952 | .968 | .970 | .973 |
| 19 | .003 | .001 | .498 | .645 | .730 | .858 | .919 | .941 | .952 | .968 | .970 | .973 |
| 20 | .003 | .001 | .497 | .644 | .730 | .858 | .918 | .941 | .951 | .967 | .970 | .973 |
| 21 | .003 | .001 | .497 | .644 | .729 | .858 | .918 | .941 | .951 | .967 | .970 | .973 |
| 22 | .003 | .001 | .497 | .644 | .729 | .857 | .918 | .940 | .951 | .967 | .970 | .973 |
| 23 | .003 | .001 | .497 | .644 | .729 | .857 | .918 | .940 | .951 | .967 | .969 | .973 |

**Similarity of Induced Weight $\hat{w}$ (Transformers v.s. Gradient Descent)**

Trasformer Layer Index

| Gradient descent Steps | 1 | 2 | 3 | 4 | 5 | 6 | 7 | 8 | 9 | 10 | 11 | 12 |
|---|---|---|---|---|---|---|---|---|---|---|---|---|
| 1 | .001 | .119 | .522 | .684 | .689 | .702 | .688 | .680 | .673 | .669 | .668 | .669 |
| 50 | .003 | .020 | .517 | .675 | .758 | .885 | .936 | .951 | .955 | .961 | .961 | .963 |
| 100 | .002 | .019 | .508 | .662 | .746 | .876 | .933 | .952 | .959 | .968 | .968 | .970 |
| 150 | .003 | .019 | .503 | .657 | .741 | .871 | .930 | .952 | .959 | .970 | .971 | .973 |
| 200 | .003 | .019 | .502 | .655 | .739 | .869 | .928 | .951 | .959 | .971 | .972 | .974 |
| 250 | .003 | .019 | .501 | .653 | .737 | .867 | .927 | .950 | .959 | .971 | .972 | .975 |
| 300 | .002 | .019 | .500 | .652 | .736 | .866 | .927 | .950 | .958 | .971 | .972 | .975 |
| 350 | .003 | .019 | .499 | .652 | .735 | .865 | .926 | .950 | .958 | .972 | .973 | .976 |
| 400 | .003 | .019 | .499 | .651 | .735 | .865 | .925 | .949 | .958 | .972 | .973 | .976 |
| 450 | .003 | .019 | .498 | .651 | .734 | .864 | .925 | .949 | .958 | .972 | .973 | .976 |
| 500 | .003 | .019 | .498 | .650 | .734 | .864 | .925 | .949 | .958 | .972 | .973 | .976 |
| 550 | .003 | .019 | .498 | .650 | .734 | .863 | .924 | .948 | .957 | .972 | .973 | .976 |
| 600 | .003 | .019 | .498 | .650 | .733 | .863 | .924 | .948 | .957 | .972 | .973 | .976 |
| 650 | .002 | .019 | .498 | .650 | .733 | .863 | .924 | .948 | .957 | .972 | .973 | .977 |
| 700 | .003 | .019 | .497 | .649 | .733 | .863 | .924 | .948 | .957 | .972 | .973 | .977 |
| 750 | .003 | .019 | .497 | .649 | .733 | .862 | .923 | .948 | .957 | .972 | .973 | .977 |
| 800 | .003 | .019 | .497 | .649 | .733 | .862 | .923 | .948 | .957 | .972 | .973 | .977 |
| 850 | .003 | .019 | .497 | .649 | .733 | .862 | .923 | .948 | .957 | .972 | .973 | .977 |
| 900 | .003 | .019 | .497 | .649 | .732 | .862 | .923 | .948 | .957 | .972 | .973 | .977 |
| 950 | .003 | .019 | .497 | .649 | .732 | .862 | .923 | .947 | .957 | .972 | .973 | .977 |
| 1000 | .003 | .019 | .497 | .649 | .732 | .862 | .923 | .947 | .957 | .972 | .973 | .977 |
| 1050 | .003 | .019 | .497 | .649 | .732 | .862 | .922 | .947 | .957 | .972 | .973 | .977 |
| 1100 | .003 | .019 | .497 | .649 | .732 | .862 | .923 | .947 | .957 | .972 | .973 | .977 |
| 1150 | .003 | .019 | .497 | .649 | .732 | .862 | .923 | .947 | .956 | .972 | .973 | .977 |
| 1200 | .003 | .019 | .496 | .648 | .732 | .861 | .922 | .947 | .956 | .972 | .973 | .977 |
| 1250 | .003 | .019 | .497 | .648 | .732 | .861 | .922 | .947 | .956 | .972 | .973 | .977 |
| 1300 | .003 | .019 | .496 | .648 | .732 | .861 | .922 | .947 | .956 | .972 | .973 | .977 |

Figure 15: **Similarity of Induced Weights on an alternative Transformers Configuration.** The best matching hyper-parameters are highlighted in yellow.

We conclude that our experimental results are not restricted to a specific model configurations, smaller models such as GPT-2 with 12 layers and 1 head each layer also suffice in implementing the Iterative Newton's method, and more similar than gradient descents, in terms of rate of convergence.

## A.4 Definitions for Evaluating Forgetting

We measure the phenomenon of model forgetting by reusing an in-context example within $\{x_i, y_i\}_{i=1}^{n}$ as the test example $x_{\text{test}}$. In experiments of Figure 5, we fix $n = 20$ and reuse $x_{\text{test}} = x_i$. We denote the "Time Stamp Gap" as the distance the reused example index $i$ from the current time stamp $n = 20$. We measure the forgetting of index $i$ as

$$\text{Forgetting}(\mathcal{A}, i) = \mathop{\mathbb{E}}_{\{x_i, y_i\}_{i=1}^{n} \sim P_{\mathcal{D}}} \text{MSE}\Big(\mathcal{A}(x_i \mid \{x_i, y_i\}_{i=1}^{n}), y_i\Big) \tag{16}$$

Note: the further away $i$ is from $n$, the more possible algorithm $\mathcal{A}$ forgets.

## B Detailed Proofs for Section 5

In this section, we work on full attention layers with normalized ReLU activation $\sigma(\cdot) = \frac{1}{n}\text{ReLU}(\cdot)$ given $n$ examples.

**Definition 5.** *A full attention layer with $M$ heads and ReLU activation is also denoted as* $\text{Attn}$ *on any input sequence* $\boldsymbol{H} = [\boldsymbol{h}_1, \cdots, \boldsymbol{h}_N] \in \mathbb{R}^{D \times N}$, *where $D$ is the dimension of hidden states and $N$ is the sequence length. In the vector form,*

$$\tilde{\boldsymbol{h}}_t = [\text{Attn}(\boldsymbol{H})]_t = \boldsymbol{h}_t + \frac{1}{n} \sum_{m=1}^{M} \sum_{j=1}^{n} \text{ReLU}\left(\langle \boldsymbol{Q}_m \boldsymbol{h}_t, \boldsymbol{K}_m \boldsymbol{h}_j \rangle\right) \cdot \boldsymbol{V}_m \boldsymbol{h}_j \tag{17}$$

**Remark 1.** *This is slightly different from the **causal** attention layer (see definition 1) in that at each time stamp $t$, the attention layer in definition 5 has full information of all hidden states $j \in [n]$, unlike causal attention layer which requires $j \in [t]$.*

## B.1 Helper Results

We begin by constructing a useful component for our proof, and state some existing constructions from Akyürek et al. (2022).

**Lemma 1.** *Given hidden states $\{h_1, \cdots, h_n\}$, there exists query, key and value matrices $Q, K, V$ respectively such that one attention layer can compute $\sum_{j=1}^{n} h_j$.*

*Proof.* We can pad each hidden state by 1 and 0's such that $h_t' \leftarrow \begin{bmatrix} h_t \\ 1 \\ 0_d \end{bmatrix} \in \mathbb{R}^{2d+1}$ . We construct two heads where $Q_1 = K_1 = Q_2 = \begin{bmatrix} O_{d \times d} & O_{d \times 1} & O_{d \times d} \\ O_{1 \times d} & 1 & O_{1 \times d} \\ O_{d \times d} & O_{d \times 1} & O_{d \times d} \end{bmatrix}$ and $K_2 = -K_1$. Then

$$\begin{bmatrix} O_{d \times d} & O_{d \times 1} & O_{d \times d} \\ O_{1 \times d} & 1 & O_{1 \times d} \\ O_{d \times d} & O_{d \times 1} & O_{d \times d} \end{bmatrix} h_t' = \begin{bmatrix} 0_d \\ 1 \\ 0_d \end{bmatrix}.$$

Let $V_1 = V_2 = \begin{bmatrix} O_{(d+1) \times d} & O_{(d+1) \times (d+1)} \\ n I_{d \times d} & O_{d \times (d+1)} \end{bmatrix}$ so that $V_m \begin{bmatrix} h_j \\ 1 \\ 0_d \end{bmatrix} = \begin{bmatrix} 0_{d+1} \\ n h_j \end{bmatrix}$.

We apply one attention layer to these 1-padded hidden states and we have

$$\begin{aligned} \tilde{h}_t &= h_t' + \frac{1}{n} \sum_{m=1}^{2} \sum_{j=1}^{n} \mathrm{ReLU}\left( \langle Q_m h_t', K_m h_j' \rangle \right) \cdot V_m h_j' \\ &= h_t' + \frac{1}{n} \sum_{j=1}^{n} \left[ \mathrm{ReLU}(1) + \mathrm{ReLU}(-1) \right] \cdot \begin{bmatrix} 0_{d+1} \\ n h_j \end{bmatrix} \\ &= \begin{bmatrix} h_t \\ 1 \\ 0_d \end{bmatrix} + \begin{bmatrix} 0_{d+1} \\ \sum_{j=1}^{n} h_j \end{bmatrix} = \begin{bmatrix} h_t \\ 1 \\ \sum_{j=1}^{n} h_j \end{bmatrix} \end{aligned} \qquad (18)$$

$\square$

**Proposition 1** ([Akyürek et al., 2022]). *Each of* mov, aff, mul, div *can be implemented by a single transformer layer. These four operations are mappings $\mathbb{R}^{D \times N} \to \mathbb{R}^{D \times N}$, expressed as follows,*

mov($H; s, t, i, j, i', j'$): *selects the entries of the $s$-th column of $H$ between rows $i$ and $j$, and copies them into the $t$-th column ($t \geq s$) of $H$ between rows $i'$ and $j'$.*

mul($H; a, b, c, (i, j), (i', j'), (i'', j'')$): *in each column $h$ of $H$, interprets the entries between $i$ and $j$ as an $a \times b$ matrix $A_1$, and the entries between $i'$ and $j'$ as a $b \times c$ matrix $A_2$, multiplies these matrices together, and stores the result between rows $i''$ and $j''$, yielding a matrix in which each column has the form $[h_{:i''-1}, A_1 A_2, h_{j'':}]^\top$. This allows the layer to implement inner products.*

div($H; (i, j), i', (i'', j'')$): *in each column $h$ of $H$, divides the entries between $i$ and $j$ by the absolute value of the entry at $i'$, and stores the result between rows $i''$ and $j''$, yielding a matrix in which every column has the form $[h_{:i''-1}, h_{i:j}/|h_{i'}|, h_{j'':}]^\top$.*

aff($H; (i, j), (i', j'), (i'', j''), W_1, W_2, \mathbf{b}$): *in each column $h$ of $H$, applies an affine transformation to the entries between $i$ and $j$ and $i'$ and $j'$, then stores the result between rows $i''$ and $j''$, yielding a matrix in which every column has the form $[h_{:i''-1}, W_1 h_{i:j} + W_2 h_{i':j'} + \mathbf{b}, h_{j'':}]^\top$. This allows the layer to implement subtraction by setting $W_1 = I$ and $W_2 = -I$.*

### B.2 PROOF OF THEOREM 1

**Theorem 1.** *There exist Transformer weights such that on any set of in-context examples $\{x_i, y_i\}_{i=1}^{n}$ and test point $x_{\text{test}}$, the Transformer predicts on $x_{\text{test}}$ using $x_{\text{test}}^\top \hat{w}_k^{\text{Newton}}$. Here $\hat{w}_k^{\text{Newton}}$ are the Iterative Newton updates given by $\hat{w}_k^{\text{Newton}} = M_k X^\top y$ where $M_j$ is updated as*

$$M_j = 2M_{j-1} - M_{j-1} S M_{j-1}, 1 \leq j \leq k, \quad M_0 = \alpha S,$$

*for some $\alpha > 0$ and $S = X^\top X$. The number of layers of the transformer is $\mathcal{O}(k)$ and the dimensionality of the hidden layers is $\mathcal{O}(d)$.*

*Proof.* We break the proof into parts.

**Transformers Implement Initialization $\boldsymbol{T}^{(0)} = \alpha\boldsymbol{S}$.** Given input sequence $\boldsymbol{H} := \{\boldsymbol{x}_1, \cdots, \boldsymbol{x}_n\}$, with $\boldsymbol{x}_i \in \mathbb{R}^d$, we first apply the mov operations given by Proposition 1 (similar to Akyürek et al. (2022), we show only non-zero rows when applying these operations):

$$\begin{bmatrix} \boldsymbol{x}_1 & \cdots & \boldsymbol{x}_n \end{bmatrix} \xrightarrow{\text{mov}} \begin{bmatrix} \boldsymbol{x}_1 & \cdots & \boldsymbol{x}_n \\ \boldsymbol{x}_1 & \cdots & \boldsymbol{x}_n \end{bmatrix} \tag{19}$$

We call each column after mov as $\boldsymbol{h}_j$. With an full attention layer, one can construct two heads with query and value matrices of the form $\boldsymbol{Q}_1^\top \boldsymbol{K}_1 = -\boldsymbol{Q}_2^\top \boldsymbol{K}_2 = \begin{bmatrix} \boldsymbol{I}_{d\times d} & \boldsymbol{O}_{d\times d} \\ \boldsymbol{O}_{d\times d} & \boldsymbol{O}_{d\times d} \end{bmatrix}$ such that for any $t \in [n]$, we have

$$\sum_{m=1}^{2} \text{ReLU}\left(\langle \boldsymbol{Q}_m\boldsymbol{h}_t, \boldsymbol{K}_m\boldsymbol{h}_j\rangle\right) = \text{ReLU}(\boldsymbol{x}_t^\top \boldsymbol{x}_j) + \text{ReLU}(-\boldsymbol{x}_t^\top \boldsymbol{x}_j) = \langle \boldsymbol{x}_t, \boldsymbol{x}_j\rangle \tag{20}$$

Let all value matrices $\boldsymbol{V}_m = n\alpha \begin{bmatrix} \boldsymbol{I}_{d\times d} & \boldsymbol{O}_{d\times d} \\ \boldsymbol{O}_{d\times d} & \boldsymbol{O}_{d\times d} \end{bmatrix}$ for some $\alpha \in \mathbb{R}$. Combining the skip connections, we have

$$\tilde{\boldsymbol{h}}_t = \begin{bmatrix} \boldsymbol{x}_t \\ \boldsymbol{x}_t \end{bmatrix} + \frac{1}{n}\sum_{j=1}^{n}\langle \boldsymbol{x}_t, \boldsymbol{x}_j\rangle n\alpha \begin{bmatrix} \boldsymbol{x}_j \\ \boldsymbol{0} \end{bmatrix} = \begin{bmatrix} \boldsymbol{x}_t \\ \boldsymbol{x}_t \end{bmatrix} + \begin{bmatrix} \alpha\left(\sum_{j=1}^{n}\boldsymbol{x}_j\boldsymbol{x}_j^\top\right)\boldsymbol{x}_t \\ \boldsymbol{0} \end{bmatrix} = \begin{bmatrix} \boldsymbol{x}_t + \alpha\boldsymbol{S}\boldsymbol{x}_t \\ \boldsymbol{x}_t \end{bmatrix} \tag{21}$$

Now we can use the aff operator to make subtractions and then

$$\begin{bmatrix} \boldsymbol{x}_t + \alpha\boldsymbol{S}\boldsymbol{x}_t \\ \boldsymbol{x}_t \end{bmatrix} \xrightarrow{\text{aff}} \begin{bmatrix} (\boldsymbol{x}_t + \alpha\boldsymbol{S}\boldsymbol{x}_t) - \boldsymbol{x}_t \\ \boldsymbol{x}_t \end{bmatrix} = \begin{bmatrix} \alpha\boldsymbol{S}\boldsymbol{x}_t \\ \boldsymbol{x}_t \end{bmatrix} \tag{22}$$

We call this transformed hidden states as $\boldsymbol{H}^{(0)}$ and denote $\boldsymbol{T}^{(0)} = \alpha\boldsymbol{S}$:

$$\boldsymbol{H}^{(0)} = \begin{bmatrix} \boldsymbol{h}_1^{(0)} & \cdots & \boldsymbol{h}_n^{(0)} \end{bmatrix} = \begin{bmatrix} \boldsymbol{T}^{(0)}\boldsymbol{x}_1 & \cdots & \boldsymbol{T}^{(0)}\boldsymbol{x}_n \\ \boldsymbol{x}_1 & \cdots & \boldsymbol{x}_n \end{bmatrix} \tag{23}$$

Notice that $\boldsymbol{S}$ is symmetric and thereafter $\boldsymbol{T}^{(0)}$ is also symmetric.

**Transformers implement Newton Iteration.** Let the input prompt be the same as Equation (23),

$$\boldsymbol{H}^{(0)} = \begin{bmatrix} \boldsymbol{h}_1^{(0)} & \cdots & \boldsymbol{h}_n^{(0)} \end{bmatrix} = \begin{bmatrix} \boldsymbol{T}^{(0)}\boldsymbol{x}_1 & \cdots & \boldsymbol{T}^{(0)}\boldsymbol{x}_n \\ \boldsymbol{x}_1 & \cdots & \boldsymbol{x}_n \end{bmatrix} \tag{24}$$

We claim that the $\ell$'s hidden states can be of the similar form

$$\boldsymbol{H}^{(\ell)} = \begin{bmatrix} \boldsymbol{h}_1^{(\ell)} & \cdots & \boldsymbol{h}_n^{(\ell)} \end{bmatrix} = \begin{bmatrix} \boldsymbol{T}^{(\ell)}\boldsymbol{x}_1 & \cdots & \boldsymbol{T}^{(\ell)}\boldsymbol{x}_n \\ \boldsymbol{x}_1 & \cdots & \boldsymbol{x}_n \end{bmatrix} \tag{25}$$

We prove by induction that assuming our claim is true for $\ell$, we work on $\ell + 1$:

Let $\boldsymbol{Q}_m = \tilde{\boldsymbol{Q}}_m \underbrace{\begin{bmatrix} \boldsymbol{O}_d & -\frac{n}{2}\boldsymbol{I}_d \\ \boldsymbol{O}_d & \boldsymbol{O}_d \end{bmatrix}}_{\boldsymbol{G}}, \boldsymbol{K}_m = \tilde{\boldsymbol{K}}_m \underbrace{\begin{bmatrix} \boldsymbol{I}_d & \boldsymbol{O}_d \\ \boldsymbol{O}_d & \boldsymbol{O}_d \end{bmatrix}}_{\boldsymbol{J}}$ where $\tilde{\boldsymbol{Q}}_1^\top \tilde{\boldsymbol{K}}_1 := \boldsymbol{I}, \tilde{\boldsymbol{Q}}_2^\top \tilde{\boldsymbol{K}}_2 := -\boldsymbol{I}$ and

$\boldsymbol{V}_1 = \boldsymbol{V}_2 = \underbrace{\begin{bmatrix} \boldsymbol{I}_d & \boldsymbol{O}_d \\ \boldsymbol{O}_d & \boldsymbol{O}_d \end{bmatrix}}_{\boldsymbol{J}}$. A 2-head self-attention layer, with ReLU attentions, can be written has

$$\boldsymbol{h}_t^{(\ell+1)} = [\text{Attn}(\boldsymbol{H}^{(\ell)})]_t = \boldsymbol{h}_t^{(\ell)} + \frac{1}{n}\sum_{m=1}^{2}\sum_{j=1}^{n}\text{ReLU}\left(\left\langle \boldsymbol{Q}_m\boldsymbol{h}_t^{(\ell)}, \boldsymbol{K}_m\boldsymbol{h}_j^{(\ell)}\right\rangle\right) \cdot \boldsymbol{V}_m\boldsymbol{h}_j^{(\ell)} \tag{26}$$

where

$$
\begin{aligned}
\sum_{m=1}^{2} &\mathrm{ReLU}\left(\left\langle \boldsymbol{Q}_m \boldsymbol{h}_t^{(\ell)}, \boldsymbol{K}_m \boldsymbol{h}_j^{(\ell)} \right\rangle\right) \cdot \boldsymbol{V}_m \boldsymbol{h}_j^{(\ell)} \\
&= \left[\mathrm{ReLU}\left((\boldsymbol{G}\boldsymbol{h}_t^{(\ell)})^\top \underbrace{\tilde{\boldsymbol{Q}}_1^\top \tilde{\boldsymbol{K}}_1}_{\boldsymbol{I}} (\boldsymbol{J}\boldsymbol{h}_j^{(\ell)})\right) + \mathrm{ReLU}\left((\boldsymbol{G}\boldsymbol{h}_t^{(\ell)})^\top \underbrace{\tilde{\boldsymbol{Q}}_2^\top \tilde{\boldsymbol{K}}_2}_{-\boldsymbol{I}} (\boldsymbol{J}\boldsymbol{h}_j^{(\ell)})\right)\right] \cdot (\boldsymbol{J}\boldsymbol{h}_j^{(\ell)}) \\
&= \left[\mathrm{ReLU}((\boldsymbol{G}\boldsymbol{h}_t^{(\ell)})^\top (\boldsymbol{J}\boldsymbol{h}_j^{(\ell)})) + \mathrm{ReLU}(-(\boldsymbol{G}\boldsymbol{h}_t^{(\ell)})^\top (\boldsymbol{J}\boldsymbol{h}_j^{(\ell)}))\right] \cdot (\boldsymbol{J}\boldsymbol{h}_j^{(\ell)}) \\
&= (\boldsymbol{G}\boldsymbol{h}_t^{(\ell)})^\top (\boldsymbol{J}\boldsymbol{h}_j^{(\ell)})(\boldsymbol{J}\boldsymbol{h}_j^{(\ell)}) \\
&= (\boldsymbol{J}\boldsymbol{h}_j^{(\ell)})(\boldsymbol{J}\boldsymbol{h}_j^{(\ell)})^\top (\boldsymbol{G}\boldsymbol{h}_t^{(\ell)})
\end{aligned}
\tag{27}
$$

Plug in our assumptions that $\boldsymbol{h}_j^{(\ell)} = \begin{bmatrix} \boldsymbol{T}^{(\ell)}\boldsymbol{x}_j \\ \boldsymbol{x}_j \end{bmatrix}$, we have $\boldsymbol{J}\boldsymbol{h}_j^{(\ell)} = \begin{bmatrix} \boldsymbol{T}^{(\ell)}\boldsymbol{x}_j \\ \boldsymbol{0}_d \end{bmatrix}$ and $\boldsymbol{G}\boldsymbol{h}_t^{(\ell)} = \begin{bmatrix} -\frac{n}{2}\boldsymbol{x}_t \\ \boldsymbol{0}_d \end{bmatrix}$, we have

$$
\begin{aligned}
\boldsymbol{h}_t^{(\ell+1)} &= \begin{bmatrix} \boldsymbol{T}^{(\ell)}\boldsymbol{x}_t \\ \boldsymbol{x}_t \end{bmatrix} + \frac{1}{n}\sum_{j=1}^{n} \begin{bmatrix} \boldsymbol{T}^{(\ell)}\boldsymbol{x}_j \\ \boldsymbol{0}_d \end{bmatrix} \begin{bmatrix} \boldsymbol{T}^{(\ell)}\boldsymbol{x}_j \\ \boldsymbol{0}_d \end{bmatrix}^\top \begin{bmatrix} -\frac{n}{2}\boldsymbol{x}_t \\ \boldsymbol{0}_d \end{bmatrix} \\
&= \begin{bmatrix} \boldsymbol{T}^{(\ell)}\boldsymbol{x}_t - \frac{1}{2}\sum_{j=1}^{n}(\boldsymbol{T}^{(\ell)}\boldsymbol{x}_j)(\boldsymbol{T}^{(\ell)}\boldsymbol{x}_j)^\top \boldsymbol{x}_t \\ \boldsymbol{x}_t \end{bmatrix} \\
&= \begin{bmatrix} \boldsymbol{T}^{(\ell)}\boldsymbol{x}_t - \frac{1}{2}\boldsymbol{T}^{(\ell)}\left(\sum_{j=1}^{n}\boldsymbol{x}_j\boldsymbol{x}_j^\top\right)\boldsymbol{T}^{(\ell)\top}\boldsymbol{x}_t \\ \boldsymbol{x}_t \end{bmatrix} \\
&= \begin{bmatrix} \left(\boldsymbol{T}^{(\ell)} - \frac{1}{2}\boldsymbol{T}^{(\ell)}\boldsymbol{S}\boldsymbol{T}^{(\ell)\top}\right)\boldsymbol{x}_t \\ \boldsymbol{x}_t \end{bmatrix}
\end{aligned}
\tag{28}
$$

Now we pass over an MLP layer with

$$
\boldsymbol{h}_t^{(\ell+1)} \leftarrow \boldsymbol{h}_t^{(\ell+1)} + \begin{bmatrix} \boldsymbol{I}_d & \boldsymbol{O}_d \\ \boldsymbol{O}_d & \boldsymbol{O}_d \end{bmatrix} \boldsymbol{h}_t^{(\ell+1)} = \begin{bmatrix} \left(2\boldsymbol{T}^{(\ell)} - \boldsymbol{T}^{(\ell)}\boldsymbol{S}\boldsymbol{T}^{(\ell)\top}\right)\boldsymbol{x}_t \\ \boldsymbol{x}_t \end{bmatrix}
\tag{29}
$$

Now we denote the iteration

$$
\boldsymbol{T}^{(\ell+1)} = 2\boldsymbol{T}^{(\ell)} - \boldsymbol{T}^{(\ell)}\boldsymbol{S}\boldsymbol{T}^{(\ell)\top}
\tag{30}
$$

We find that $\boldsymbol{T}^{(\ell+1)\top} = \boldsymbol{T}^{(\ell+1)}$ since $\boldsymbol{T}^{(\ell)}$ and $\boldsymbol{S}$ are both symmetric. It reduces to

$$
\boldsymbol{T}^{(\ell+1)} = 2\boldsymbol{T}^{(\ell)} - \boldsymbol{T}^{(\ell)}\boldsymbol{S}\boldsymbol{T}^{(\ell)}
\tag{31}
$$

This is exactly the same as the Newton iteration.

**Transformers can implement $\hat{\boldsymbol{w}}_\ell^{\mathrm{TF}} = \boldsymbol{T}^{(\ell)}\boldsymbol{X}^\top \boldsymbol{y}$.** Going back to the empirical prompt format $\{\boldsymbol{x}_1, y_1, \cdots, \boldsymbol{x}_n, y_n\}$. We can let parameters be zero for positions of $y$'s and only rely on the skip connection up to layer $\ell$, and the $\boldsymbol{H}^{(\ell)}$ is then $\begin{bmatrix} \boldsymbol{T}^{(\ell)}\boldsymbol{x}_j & \boldsymbol{0} \\ \boldsymbol{x}_j & \boldsymbol{0} \\ 0 & y_j \end{bmatrix}_{j=1}^n$. We again apply operations from Proposition 1:

$$
\begin{bmatrix} \boldsymbol{T}^{(\ell)}\boldsymbol{x}_j & \boldsymbol{0} \\ \boldsymbol{x}_j & \boldsymbol{0} \\ 0 & y_j \end{bmatrix}_{j=1}^n \xrightarrow{\mathrm{mov}} \begin{bmatrix} \boldsymbol{T}^{(\ell)}\boldsymbol{x}_j & \boldsymbol{T}^{(\ell)}\boldsymbol{x}_j \\ \boldsymbol{x}_j & \boldsymbol{0} \\ 0 & y_j \end{bmatrix}_{j=1}^n \xrightarrow{\mathrm{mul}} \begin{bmatrix} \boldsymbol{T}^{(\ell)}\boldsymbol{x}_j & \boldsymbol{T}^{(\ell)}\boldsymbol{x}_j \\ \boldsymbol{x}_j & \boldsymbol{0} \\ 0 & y_j \\ \boldsymbol{0} & \boldsymbol{T}^{(\ell)}y_j\boldsymbol{x}_j \end{bmatrix}_{j=1}^n
\tag{32}
$$

Now we apply Lemma 1 over all even columns in Equation (32) and we have

$$
\text{Output} = \sum_{j=1}^{n} \begin{bmatrix} \boldsymbol{T}^{(\ell)}\boldsymbol{x}_j \\ \boldsymbol{0} \\ y_j \\ \boldsymbol{T}^{(\ell)}y_j\boldsymbol{x}_j \end{bmatrix} = \begin{bmatrix} \boldsymbol{\xi} \\ \boldsymbol{T}^{(\ell)}\sum_{j=1}^{n}y_j\boldsymbol{x}_j \end{bmatrix} = \begin{bmatrix} \boldsymbol{\xi} \\ \boldsymbol{T}^{(\ell)}\boldsymbol{X}^\top\boldsymbol{y} \end{bmatrix}
\tag{33}
$$

where $\boldsymbol{\xi}$ denotes irrelevant quantities. Note that the resulting $\boldsymbol{T}^{(\ell)}\boldsymbol{X}^\top\boldsymbol{y}$ is also the same as Iterative Newton's predictor $\hat{\boldsymbol{w}}_k = \boldsymbol{M}_k\boldsymbol{X}^\top\boldsymbol{y}$ after $k$ iterations. We denote $\hat{\boldsymbol{w}}_\ell^{\mathrm{TF}} = \boldsymbol{T}^{(\ell)}\boldsymbol{X}^\top\boldsymbol{y}$.

**Transformers can make predictions on $\boldsymbol{x}_{test}$ by $\left\langle\hat{\boldsymbol{w}}_\ell^{\mathrm{TF}}, \boldsymbol{x}_{test}\right\rangle$.**

Now we can make predictions on text query $\boldsymbol{x}_{\mathrm{test}}$:

$$
\begin{bmatrix} \boldsymbol{\xi} & \boldsymbol{x}_{\mathrm{test}} \\ \hat{\boldsymbol{w}}_\ell^{\mathrm{TF}} & \boldsymbol{x}_{\mathrm{test}} \end{bmatrix} \xrightarrow{\mathrm{mov}} \begin{bmatrix} \boldsymbol{\xi} & \boldsymbol{x}_{\mathrm{test}} \\ \hat{\boldsymbol{w}}_\ell^{\mathrm{TF}} & \boldsymbol{x}_{\mathrm{test}} \\ \boldsymbol{0} & \hat{\boldsymbol{w}}_\ell^{\mathrm{TF}} \end{bmatrix} \xrightarrow{\mathrm{mul}} \begin{bmatrix} \boldsymbol{\xi} & \boldsymbol{x}_{\mathrm{test}} \\ \hat{\boldsymbol{w}}_\ell^{\mathrm{TF}} & \boldsymbol{x}_{\mathrm{test}} \\ \boldsymbol{0} & \hat{\boldsymbol{w}}_\ell^{\mathrm{TF}} \\ 0 & \left\langle\hat{\boldsymbol{w}}_\ell^{\mathrm{TF}}, \boldsymbol{x}_{\mathrm{test}}\right\rangle \end{bmatrix} \tag{34}
$$

Finally, we can have an readout layer $\boldsymbol{\beta}_{\mathrm{ReadOut}} = \{\boldsymbol{u}, v\}$ applied (see definition 3) with $\boldsymbol{u} = \begin{bmatrix} \boldsymbol{0}_{3d} & 1 \end{bmatrix}^\top$ and $v = 0$ to extract the prediction $\left\langle\hat{\boldsymbol{w}}_\ell^{\mathrm{TF}}, \boldsymbol{x}_{\mathrm{test}}\right\rangle$ at the last location, given by $\boldsymbol{x}_{\mathrm{test}}$. This is exactly how Iterative Newton makes predictions.

**To Perform $k$ steps of Newton's iterations, Transformers need $\mathcal{O}(k)$ layers.**

Let's count the layers:

- **Initialization**: `mov` needs $\mathcal{O}(1)$ layer; gathering $\alpha\boldsymbol{S}$ needs $\mathcal{O}(1)$ layer; and `aff` needs $\mathcal{O}(1)$ layer. In total, Transformers need $\mathcal{O}(1)$ layers for initialization.

- **Newton Iteration**: each exact Newton's iteration requires $\mathcal{O}(1)$ layer. Implementing $k$ iterations requires $\mathcal{O}(k)$ layers.

- **Implementing $\hat{\boldsymbol{w}}_\ell^{\mathrm{TF}}$**: We need one operation of `mov` and `mul` each, requiring $\mathcal{O}(1)$ layer each. Apply Lemma 1 for summation also requires $\mathcal{O}(1)$ layer.

- **Making prediction on test query**: We need one operation of `mov` and `mul` each, requiring $\mathcal{O}(1)$ layer each.

Hence, in total, Transformers can implement $k$-step Iterative Newton and make predictions accordingly using $\mathcal{O}(k)$ layers.

$\square$

### B.3 ITERATIVE NEWTON AS A SUM OF MOMENTS METHOD

Recall that Iterative Newton's method finds $\boldsymbol{S}^\dagger$ as follows

$$
\boldsymbol{M}_0 = \underbrace{\frac{2}{\|\boldsymbol{S}\boldsymbol{S}^\top\|_2}}_{\alpha} \boldsymbol{S}^\top, \qquad \boldsymbol{M}_k = 2\boldsymbol{M}_{k-1} - \boldsymbol{M}_{k-1}\boldsymbol{S}\boldsymbol{M}_{k-1}, \forall k \geq 1. \tag{35}
$$

We can expand the iterative equation to moments of $\boldsymbol{S}$ as follows.

$$
\boldsymbol{M}_1 = 2\boldsymbol{M}_0 - \boldsymbol{M}_0\boldsymbol{S}\boldsymbol{M}_0 = 2\alpha\boldsymbol{S}^\top - 4\alpha^2\boldsymbol{S}^\top\boldsymbol{S}\boldsymbol{S}^\top = 2\alpha\boldsymbol{S} - 4\alpha^2\boldsymbol{S}^3. \tag{36}
$$

Let's do this one more time for $\boldsymbol{M}_2$.

$$
\begin{aligned}
\boldsymbol{M}_2 &= 2\boldsymbol{M}_1 - \boldsymbol{M}_1\boldsymbol{S}\boldsymbol{M}_1 \\
&= 2(2\alpha\boldsymbol{S} - 4\alpha^2\boldsymbol{S}^3) - (2\alpha\boldsymbol{S} - 4\alpha^2\boldsymbol{S}^3)\boldsymbol{S}(2\alpha\boldsymbol{S} - 4\alpha^2\boldsymbol{S}^3) \\
&= 4\alpha\boldsymbol{S} - 8\alpha^2\boldsymbol{S}^3 - 4\alpha^2\boldsymbol{S}^3 + 16\alpha^3\boldsymbol{S}^5 - 16\alpha^4\boldsymbol{S}^7 \\
&= 4\alpha\boldsymbol{S} - 12\alpha^2\boldsymbol{S}^3 + 16\alpha^3\boldsymbol{S}^5 - 16\alpha^4\boldsymbol{S}^7.
\end{aligned} \tag{37}
$$

We can see that $\boldsymbol{M}_k$ are summations of moments of $\boldsymbol{S}$, with respect to some pre-defined coefficients from the Newton's algorithm. Hence Iterative Newton is a special of an algorithm which computes an approximation of the inverse using higher-order moments of the matrix,

$$
\boldsymbol{M}_k = \sum_{s=1}^{2^{k+1}-1} \beta_s\boldsymbol{S}^s \tag{38}
$$

with coefficients $\beta_s \in \mathbb{R}$.

We note that Transformer circuits can represent other sum of moments other than Newton's method. We can introduce different coefficients $\beta_i$ than in the proof of Theorem 1 by scaling the value matrices or through the MLP layers.

### B.4 ESTIMATED WEIGHT VECTORS LIE IN THE SPAN OF PREVIOUS EXAMPLES

What properties can we infer and verify for the weight vectors which arise from Newton's method? A straightforward one arises from interpreting any sum of moments method as a kernel method.

We can expand $\boldsymbol{S}^s$ as follows

$$\boldsymbol{S}^s = \left(\sum_{i=1}^{t} \boldsymbol{x}_i \boldsymbol{x}_i^\top\right)^s = \sum_{i=1}^{t}\left(\sum_{j_1,\cdots,j_{s-1}} \langle \boldsymbol{x}_i, \boldsymbol{x}_{j_1}\rangle \prod_{v=1}^{s-2}\langle \boldsymbol{x}_{j_v}, \boldsymbol{x}_{j_{v+1}}\rangle\right)\boldsymbol{x}_i \boldsymbol{x}_{j_{s-1}}^\top. \tag{39}$$

Then we have

$$
\begin{aligned}
\hat{\boldsymbol{w}}_t = \boldsymbol{M}_t \boldsymbol{X}^\top \boldsymbol{y} &= \sum_{s=1}^{2^{t+1}-1} \beta_s \boldsymbol{S}^s \boldsymbol{X}^\top \boldsymbol{y} \\
&= \sum_{s=1}^{2^{t+1}-1} \beta_s \left\{\sum_{i=1}^{t}\left(\sum_{j_1,\cdots,j_{s-1}}\langle \boldsymbol{x}_i,\boldsymbol{x}_{j_1}\rangle \prod_{v=1}^{s-2}\langle \boldsymbol{x}_{j_v},\boldsymbol{x}_{j_{v+1}}\rangle\right)\boldsymbol{x}_i \boldsymbol{x}_{j_{s-1}}^\top\right\}\left\{\sum_{i=1}^{t} y_i \boldsymbol{x}_i\right\} \\
&= \sum_{s=1}^{2^{t+1}-1} \beta_s \left(\sum_{i=1}^{t}\left(\sum_{j_1,\cdots,j_s} y_{j_s}\langle \boldsymbol{x}_i,\boldsymbol{x}_{j_1}\rangle \prod_{v=1}^{s-1}\langle \boldsymbol{x}_{j_v},\boldsymbol{x}_{j_{v+1}}\rangle\right)\boldsymbol{x}_i\right) \\
&= \sum_{i=1}^{t}\underbrace{\left(\sum_{s=1}^{2^{t+1}-1}\sum_{j_1,\cdots,j_s}\beta_s y_{j_s}\langle \boldsymbol{x}_i,\boldsymbol{x}_{j_1}\rangle \prod_{v=1}^{s-1}\langle \boldsymbol{x}_{j_v},\boldsymbol{x}_{j_{v+1}}\rangle\right)}_{\phi_t(i|\boldsymbol{X},\boldsymbol{y},\boldsymbol{\beta})}\boldsymbol{x}_i \\
&= \sum_{i=1}^{t}\phi_t(i \mid \boldsymbol{X},\boldsymbol{y},\boldsymbol{\beta})\,\boldsymbol{x}_i
\end{aligned} \tag{40}
$$

where $\boldsymbol{X}$ is the data matrix, $\boldsymbol{\beta}$ are coefficients of moments given by the sum of moments method and $\phi_t(\cdot)$ is some function which assigns some weight to the $i$-th datapoint, based on all other datapoints. Therefore if the Transformer implements a sum of moments method (such as Newton's method), then its induced weight vector $\tilde{\boldsymbol{w}}_t(\text{Transformers} \mid \{\boldsymbol{x}_i, y_i\}_{i=1}^{t})$ after seeing in-context examples $\{\boldsymbol{x}_i, y_i\}_{i=1}^{t}$ should lie in the span of the examples $\{\boldsymbol{x}_i\}_{i=1}^{t}$:

$$\tilde{\boldsymbol{w}}_t(\text{Transformers} \mid \{\boldsymbol{x}_i, y_i\}_{i=1}^{t}) \overset{?}{=} \text{Span}\{\boldsymbol{x}_1,\cdots,\boldsymbol{x}_t\} = \sum_{t=1}^{t} a_i \boldsymbol{x}_i \qquad \text{for coefficients } a_i. \tag{41}$$

We test this hypothesis. Given a sequence of in-context examples $\{\boldsymbol{x}_i, y_i\}_{i=1}^{t}$, we fit coefficients $\{a_i\}_{i=1}^{t}$ in Equation (41) to minimize MSE loss:

$$\{\hat{a}_i\}_{i=1}^{t} = \underset{a_1,a_2,\cdots,a_t \in \mathbb{R}}{\arg\min}\left\|\tilde{\boldsymbol{w}}_t(\text{Transformers} \mid \{\boldsymbol{x}_i, y_i\}_{i=1}^{t}) - \sum_{t=1}^{t} a_i \boldsymbol{x}_i\right\|_2^2. \tag{42}$$

We then measure the quality of this fit across different number of in-context examples $t$, and visualize the residual error in Figure 16. We find that even when $t < d$, Transformers' induced weights still lie close to the span of the observed examples $\boldsymbol{x}_i$'s. This provides an additional validation of our proposed mechanism.

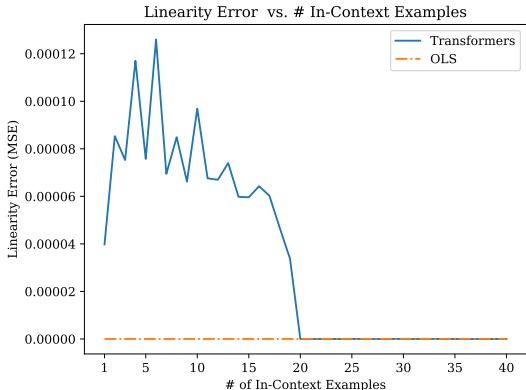

Figure 16: Verification of hypothesis that the Transformers induced weight vector $w$ lies in the span of observed examples $\{x_i\}$.

