# OpenReview forum: "Transformers Learn Higher-Order Optimization Methods for In-Context Learning: A Study with Linear Models"
_ICLR.cc/2024/Conference — Submitted to ICLR 2024_

### Official Review · Reviewer_TsNZ · 2023-10-25

**Soundness:** 2 fair
**Presentation:** 3 good
**Contribution:** 2 fair
**Rating:** 3
**Confidence:** 3

**Summary:**

This paper focuses on the algorithmic side of in-context learning. The authors claim that the transformer implements ICL via the iterative Newton’s method. The paper provides numerical comparisons among the behaviors of ICL, GD, and iterative Newton’s method. The results show that the ICL is more similar to iterative Newton’s method. The authors also provide the approximation results to support this claim.

**Strengths:**

This paper adopts different metrics to indicate that the algorithm implemented by ICL is closer to iterative Newton’s method than GD. In addition, the ill-conditioned setting is also included to demonstrate this similarity. The theoretical results from the approximation view are provided to verify the claim in the paper.

**Weaknesses:**

Some of my concerns are not addressed after reading this paper:

1. It is unclear why the similarity of errors is a reasonable metric. It would be helpful if the paper could provide more clarity on why the $l_2$ norm is not directly used to measure the distance between predictions instead of the error. In addition, the scale of the error is important to the problem in the linear regression problem. Since normalizing the error is mentioned in the paper, it would be beneficial to provide a more detailed explanation of why this normalization is necessary.

2. I have a query about the reasonability of the definition of induced weights for the LLM. Concretely, given any examples in the prompt, if we scale the $x_{query}$ by a constant, will the output be scaled by the same constant? If not, it may be improper to define the induced weights for the LLM. More numerical verifications of this should be provided. This verification is essential since the softmax module is known to be a strongly non-convex function.

3. There is some uncertainty about using the **best** hyperparameter to measure the similarity between algorithms as discussed in the paper. For example, we can simply define a new algorithm as a linear combination of the iterative Newton’s method and GD. Trivially, this algorithm with the best hyperparameters will fit ICL of LLM better than iterative Newton’s method, since iterative Newton’s method is this new algorithm with the combination coefficient as $1$. Obviously, such a comparison is meaningless. The main reason is that setting the best hyperparameter gives too much **freedom** to the algorithm. Thus, it is very helpful to discuss why we can use the best hyperparameter to compare the algorithms in the setting of this paper.

4. It will be beneficial to discuss whether it is possible to approximate the iterative Newton’s method with causal attention. It will better align the theoretical results and the empirical findings.

**Questions:**

The questions are listed in the weakness section.

---

> ### Author Response · Authors · 2023-11-17
>
> We thank the reviewer for providing constructive feedback.
>
> **The reviewer asked why we use cosine similarity on errors.** Cosine similarity is better for cases where values are small. Usually errors (also known as residuals) have quite small magnitudes. Directly measuring the prediction will usually yield high similarities. We also ran comparisons using the L2 norm and didn’t find a difference in convergence rates. Our alternative metric (similarity of induced weights) also agrees with our cosine similarity metrics.
>
> **The reviewer questioned the validity of using Induced Weights.** Samples used to induce these weights are coming from the same distribution as our in-context examples. As Transformers can make good predictions on $x$ sampled in-distribution, and the true $x \rightarrow y$ mapping is linear, it’s reasonable to approximate Transformers with a linear function too. The induced weights are calculated through solving a deterministic linear system. We wouldn't think of out-of-distribution samples (for example scaling) at this stage. This is also done by [2] in Eqn. 16 in section 4.1.
>
> [2] Ekin Akyürek, Dale Schuurmans, Jacob Andreas, Tengyu Ma, and Denny Zhou. *What learning algorithm is in-context learning? investigations with linear models.*
>
> **The reviewer questioned using best-matching hyperparameters to compare algorithms.** The hyperparameters we match for comparisons are the number of steps of GD/Newton v.s. the number of layers of Transformers. The intuition behind is: how many more steps of Newton or GD are needed to match the performance of an additional layer of Transformers. Afterwards, we can use this to compare the rate of convergence between algorithms, and we show that Transformers and Iterative Newton can have the same convergence rate and both are exponentially faster than GD.
>
> **The reviewer asked whether we can use Causal Attention for theory.** It can be relaxed if we only care about cases when the number of in-context examples are greater than the data dimension. Under such assumption, in Eqn. 32, $\frac{1}{n} \sum_{j=1}^n x_j x_j^\top$  will become $\frac{1}{t} \sum_{j=1}^t x_j x_j^\top$, when $t \geq d$, this is a valid estimator of the covariance matrix $\hat{\Sigma}$ and the rest of the proof remain similar. However, for $t < d$, this is not a valid estimator any more.

---

> ### Comment · Reviewer_TsNZ · 2023-11-23
>
> Thank the authors for the detailed response. My concerns regarding the performance metric and causal attention are well addressed. It is encouraged to also include the simulation results about $l _2$ loss in the main paper or at least mention them.
>
> My concern about the best-matching parameters still remains. The overfitting may appear due to the "best-matching". For example, a naive algorithm is an ergodic random walk on the subset that the results of GD and transformers belong to. Then we can always find a step of this naive algorithm that is close to the results of transformers. Thus, I will maintain my scores.

---

> ### Author Response · Authors · 2023-11-23
> **Followup Discussion with Reviewer TsNZ**
>
> We thank the reviewer for discussions.
>
> In the experiments, there are two sets of hyper-parameters:
>
> > 1. Hyper-parameters specific to algorithms (learning rate of GD and initialization scale for Newton): these hyper-parameters are determined before running the algorithms. As long as they are chosen to guarantee convergence, they won’t affect the **rate of convergence**.
> > 2. Hyper-parameters that determine the rate of convergence: number of steps of GD, number of steps of Iterative Newton, and number of layers of Transformers.
>
> In comparing the rate of convergence, we compare the second set of hyper-parameters (steps/iterations). In order to compare which algorithm convergences faster, we need do the following: Fixing error rate $\epsilon$, if Transformers can achieve $\epsilon$ error with $\ell$ layers, how many steps of Iterative Newton or GD are needed to achieve the same $\epsilon$ error. We find that Iterative Newton needs $\mathcal O(\ell)$ steps and GD needs $\mathcal O(\exp(\ell))$ steps. Our experiments are designed accordingly to compare this.
>
> We acknowledge the confusion of using the same name “hyper-parameters” to address two sets of parameters and we’ll modify the terminology of the second to be “best-matching steps for comparing convergence”.
>
> **The reviewer also mentioned that “linearly combining GD and Newton” solutions can produce a better algorithm than Newton**. This is not true. Ignoring the $\kappa$ term in well-conditioned cases, Newton converges to $\epsilon$ error with $\mathcal O(\log \log (1/\epsilon))$ steps, while GD is exponentially slower with $\mathcal O(\log (1/\epsilon))$ steps. Combining the two may produce faster algorithms than GD but any interpolation of a fast algorithm (Newton) and a slow algorithm (GD) *cannot* produce a faster algorithm than the fast one (Newton).

---

### Official Review · Reviewer_zXY7 · 2023-11-01

**Soundness:** 2 fair
**Presentation:** 2 fair
**Contribution:** 3 good
**Rating:** 3
**Confidence:** 2

**Summary:**

This paper investigates how transformers solve linear regression problems, providing a novel perspective compared to previous work which demonstrated that transformers could implement a gradient descent update at each layer. In contrast, this paper empirically demonstrates that transformers tend to implement an algorithm more akin to the iterative Newton's method. Furthermore, the authors present theoretical findings that prove the existence of transformer constructions capable of performing Newton's method.

**Strengths:**

1. The paper introduces two distinct metrics for measuring the similarity between different algorithms applied to solving linear regression problems, contributing valuable tools for future research.
2. A comprehensive set of experiments is conducted to delve into the relationships between Newton's methods, gradient descent, Transformers, online gradient descent, and LSTMs. The findings indicate that Transformers approximate the iterative Newton's method, whereas LSTMs align more closely with online gradient descent.
3. The authors provide a proof, demonstrating the existence of a Transformer architecture with $O(k)$ layers that can perform $k$ iterations of Newton's method on given examples, which supports their empirical findings.

**Weaknesses:**

1. The manuscript appears to have been submitted without thorough proofreading, as it contains numerous typographical and grammatical errors.
2. The paper lacks crucial experimental details. While some information is provided, such as data distribution, transformer architecture, and in-context length, many important implementation specifics are omitted. For instance, the paper does not disclose the learning rates used for the iterative Newton's method and gradient descent, the criteria for selecting $T$ in relation to $d$ for calculating similarity of induced weights, or the methodology employed to determine the best matched hyperparameters as per Definition 4.
3. The depiction in Figure 3 raises concerns. Given a sufficient sample size and a noise-free data setting, all methods—Newton's method, gradient descent, and transformers—should empirically converge to a correct prediction. Intuitively, this should result in a similarity score of $0.99$ between the final output of the transformer (after the $L$th layer) and the other two methods. However, this is not reflected in Figure 3's right lower corner. This discrepancy prompts further investigation: could additional training iterations, an optimized learning rate, or other modifications enhance the gradient descent results, enabling it to make precise predictions akin to the other methods?

**Questions:**

Please refer to the Weaknesses section for potential areas of inquiry and clarification.

---

> ### Author Response · Authors · 2023-11-17
>
> We thank the reviewer for providing constructive feedback.
>
> **The reviewer asked for details about hyperparameters.** Iterative Newton has no learning rate. We choose the best learning rate for GD via grid search to ensure convergence, and here it’s 0.01. As long as GD converges, it won’t affect our claim on the rate of convergence. The best matching hyperparameters (number of steps in GD/Newton v.s. number of layers in Transformers) are selected by taking the argmax of similarity.
>
> **The reviewer pointed out GD should converge to the OLS.** This is indeed the case, however it converges much more slowly. We have thus updated our claims to be about convergence rates. we revised our experiments so that all three methods of interest converge to the OLS solution and the major difference is the rate of convergence.

---

### Official Review · Reviewer_CBs3 · 2023-11-02

**Soundness:** 3 good
**Presentation:** 3 good
**Contribution:** 3 good
**Rating:** 6
**Confidence:** 3

**Summary:**

This paper performs numerical simulations to show that transformers learn to implement an algorithm very similar to Iterative Newton's Method rather than Gradient Descent. They also show that Transformers can learn in context on ill-conditioned data and theoretically show that transformers can implement k iterations of Newton's method with O(k) layers.

**Strengths:**

1. The papers perform numerical simulations to demonstrate that the algorithms implemented by transformers are more similar to Newton's method. The experimental results are interesting.
2. The paper is written clearly.

**Weaknesses:**

1. This paper only compared two optimization methods: Newton's method and gradient descent (with a particular step size?). It is very straightforward to see that transformers can implement other first-order methods, including accelerated gradient descent and approximate message passing. It is not clear if there could be another first-order method that is more similar to Newton's method.
2. Construction that transformers can implement Newton's method exists in [1] Section 8.

[1] Looped Transformers as Programmable Computers. Angeliki Giannou, Shashank Rajput, Jy-yong Sohn, Kangwook Lee, Jason D. Lee, Dimitris Papailiopoulos.

**Questions:**

1. What is the step size for GD in experiments? Is the result robust to the tuning of step size?
2. Why, in Figure 3, do Newton's method and Transformer layer 12 have a similarity of 0.994, but GD and Transformer only have a similarity of 0.88? Intuitively, GD and Newton's method should both converge to the minimum-norm OLS solution. In Figure 9, it seems to say that the CosSim of w_{GD} and w_{OLS} is small when the number of in-context examples is less than 20. However, people have proved that GD starting from zero initialization should converge to the min-norm least square solution. Why is there a mismatch between theory and your experiment?

---

> ### Author Response · Authors · 2023-11-17
>
> We thank the reviewer for providing constructive feedback.
>
> **The reviewer asked whether another first-order method can be more similar to Newton's method.** In terms of rate of convergence, the lower bound for gradient-based methods is $\Omega(\sqrt{\kappa(S)} \log(1/\epsilon))$ [1]. It is still exponentially slower than Newton’s method.
>
> [1] A.S. Nemirovski and D.B Yudin. *Problem complexity and method efficiency in optimization.*
>
> **The reviewer pointed out a missing reference on Looped Transformers that can implement matrix inverse by Iterative Newton’s method.** We thank the reviewer for pointing it out. We believe their construction could be used as a subroutine in an alternative proof of our claim, however one major difference is that our construction explicitly considers regression tasks whereas their construction concerns matrix inversion. Our construction never explicitly calculates the Newton inverse $M_k$ (at step $k$): we always consider $M_k x_j$ for some input token $x_j$. Replacing our construction with looped transformers also requires additional circuits to compute $X^\top y$ and make predictions using $\langle M_k X^\top y, x_\mathrm{test}\rangle$. We will discuss more about this in our next revision.
>
> **Question 1: The reviewer asked about the step size of GD.** We choose the best learning rate for GD via grid search to ensure convergence, and here it’s 0.01. As long as GD converges, it won’t affect our claim on the rate of convergence.
>
> **Question 2: The reviewer pointed out GD should converge to the OLS.** This is indeed the case, however it converges much more slowly. We have thus updated our claims to be about convergence rates. we revised our experiments so that all three methods of interest converge to the OLS solution and the major difference is the rate of convergence.

---

### Official Review · Reviewer_eSFG · 2023-11-11

**Soundness:** 2 fair
**Presentation:** 2 fair
**Contribution:** 2 fair
**Rating:** 5
**Confidence:** 3

**Summary:**

This paper showed that trained Transformers can implement some higher-order optimization method like Newton method. They measured  the algorithmic similarity based on the cosine difference of the errors and the induced weight, and showed that Transformers are more  similar to Newton's iterate rather than vanilla gradient descent. They showed that roughly one-layer of TF can  implement three steps of Newton's step on linear regression problem. They also provide theoretical guarantee that bi-directional attention model with ReLU activation function can implement k-steps of Newton's iterate using O(k) layers.

Generally speaking, this is an interesting work, but the result is not too solid. I will explain it in the weakness section. It is likely for me to raise my score given the rebuttal.

**Strengths:**

1. The insight that Transformers can implement higher order optimization methods which are superior to vanilla gradient descent is very interesting topic and worth investigating in the future. The motivation that viewing transformer as some meta-optimizer is very promising.

2.  The visualization is clear and the theory looks good. They proved that O(k) layers of Transformers can implement k-steps of Newton's iterates, which is good.

3. They considered the comparison to LSTM, as well as testing on the ill-conditioned linear regression case, which solidify this paper a lot.

4. The literature review is good and the writing is clear and easy to follow.

**Weaknesses:**

1. The greatest issue is that there is gaps between the theoretical results and experiments results, so it is not clear **what algorithm does TF exactly implement?** The author proves that the TF can implements k-steps of Newton's iterates using O(k) layers (at least k layers, right?) But the experiments showed that TF can approximate 3 Newton's step using one single layer. This suggests that, probably, the trained TF ca do something smarter than Newton's step, and this is what we care about the most: can we discover some new or more efficient algorithms by Transformers?

I know this is a difficult question, so I do not expect the authors to answer it thoroughly, but I think it will be better to have an explanation about 'why one layer of TF can implement 3 steps of Newton's step'? If you can provide some intuition about  how does it exactly achieve this, this will definitely be a perfect result and improve this paper by a lot.

**Questions:**

1. The second question is about figure 3 and figure 7. Since the main claim in your paper is 'TF are more similar to Newton's iterate than vanilla gradient descent' and your main evidence is these two figures, we need to examine these two papers carefully.

1.1 In figure 7 for example, in column 8, there are 4 items with the same value (.988), why do you only highlight one of them? How do you choose it? Since you claimed that from 3th-9th layer, there is a linear trend for the number of Newton's iterate versus the number of TF layer, it matters which items you highlight. For example, if you highlight some other entries with the same value inside, this will not look like a linear trend at all. Similarly, in figure7 right, there are many 0.884 in the 7th column, so why do you highlight one of them?

1.2 In both subfigure of figure 7, there are multiple items with very similar values. For example, in the column 9 on the left, there are .992, 0.991, 0.990, 0.993. Since you chose the highest value in each column, then one natural question is, is this choice statistically significant? Did you do the experiment for multiple times? Using different random seeds, will the items you chose change or not? I think since the values in grids are too closed, verifying the significance is critical.

1.3 How do you tune the step size of the vanilla GD method and Newton's iterate? This is critical since different step sizes correspond to totally different learning algorithms.

1.4 In principle, both Newton's method and vanilla GD will converge to OLS, but why the maximal similarity between TF and Newton is larger than the maximal similarity between TF and GD? For example, in figure 7, the maximal similarity between TF and Newton is 0.974, while the GD is 0.907, which is strange, right? With enough steps, both GD and Newton should approximately equal to OLS, so the maximal similarity value between  TF and GD (Newton) should be the same, when GD and Newton converge. Why are they different in your experiments?

2. About LSTM:

2.1 Why do you claim 'LSTM is more  similar to OGD than to Newton's method'? In the fig5 left, this shows that LSTM is more similar to '5 steps of Newton', while in the fig5 center, the LSTM curve is closer to that of OGD. Comparing these two, how do you compare which one is more similar to LSTM? Why don't you claim that LSTM is more similar to '5-step of Newton' based on the left figure?

2.2 There is some issue with Table1. Your claim is 'LSTM is more similar to OGD than to Newton/GD'. In order to prove this, you need to compare the three numbers in the right column: 0.920, 0.808, .831. Based on this, actually LSTM is more similar to Newton than to OGD. In your paper, you highlight the larger number in each row, which actually means 'OGD is more similar to TF than to LSTM'. But this does not match your main claim. Do you have any explanation?

3. About the theory: There is one issue for Theorem 1. You claim that 'there is a TF such that for any set of in-context sample ......'. This can not be true, since Newton's iteration require some condition on the \alpha (in your definition of M_0) to converge. Basically, for each set of in-context example, even if your TF construction implement the Newton's iterate, you still need to choose the proper \alpha to make this iterate converge. The proper choice of \alpha should be depend on eigenvalues of X^\top X, and the Transformer you construct should depend on this \alpha, which again depend on your dataset {x_i, y_i}.

4. Some small questions:

4.1 In section 3, you first sample a covariance matrix \Sigma from some distribution, and then sample feature vectors for each task, but in most of your results, you fix this covariance matrix. I suggest that you should build your set up based on this fixed covariance case, since basically training on fixed cov data and random cov data will be very different.

4.2 In measuring the similarity between errors and induced weight, why do you use cosine metric instead of common L2 norm? is there a particular reason to use cosine? I think if your goal is to claim that TF is mimicking some existing algorithm, why not use L2 norm? A small cosine value only indicates the closeness in direction, which will weaken your arguments.

**Details Of Ethics Concerns:**

/

---

> ### Author Response · Authors · 2023-11-17
>
> We thank the reviewer for providing constructive feedback.
>
> **The reviewer asked whether trained Transformers can do something smarter than Newton.** We agree this is possible. As discussed in Section 5 and Appendix B.3, we showed that Iterative Newton is a specific version of the “sum of moments” method. Trained Transformer is also such a method, but could have better coefficients. Nevertheless, this won’t change the asymptotic rate of convergence (measured as big-O), since Newton’s method matches the lower bound for the problem.
>
> **Question 1: The reviewer pointed out GD should converge to the OLS.** This is indeed the case, however it converges much more slowly. We have thus updated our claims to be about convergence rates. In our revised manuscript, we fix this issue: GD, Iterative Newton, and Transformers will all converge to the OLS solution and the major difference is the rate of convergence. GD converges exponentially slower than the other two. The reviewer asked for learning rates used for GD. We choose the best learning rate for GD to ensure convergence, and here it’s 0.01 in well-conditioned cases. As long as GD converges, it won’t affect our claim on the rate of convergence.
>
> **Question 2: The reviewer asked whether LSTM is also more similar to Newton with 5 steps.** We also updated the OGD experiments by initializing w to zero (previously it was initialized randomly). In the new Fig. 5, LSTM matches well with both OGD and 5-step Newton in terms of in-context performance. However, LSTM and OGD have the same level of forgetting issues but Newton does not. In the updated table, even compared vertically, LSTM matches OGD the most.
>
> **Question 3: The reviewer asked how Transformers can decide $\alpha$ which can depend on $S S^\top$ in Eqn. 8.** For Iterative Newton’s method, $\alpha$ can be any value between 0 and  $\frac{2}{||S S^\top||_2}$  to ensure convergence. We pick $\frac{2}{||S S^\top||_2}$ in experiments in order to reduce randomness. Transformers can learn any fixed $\alpha$ as long as it is sufficiently small.
>
> **Question 4.1: The reviewer suggested we should describe $\Sigma$ as fixed in problem setup.** In most of the experiments we fix $\Sigma$ as identity (see the ending sentence at the first paragraph of Sec. 3). We also need to sample $\Sigma$ for ill-conditioned problems (so that the eigenbasis is randomly sampled to prevent Transformers from being lazy by memorizing a fixed $\Sigma$), where it further verifies that both Transformers and Newton won’t suffer from large condition number but gradient-based methods do.
>
> **Question 4.2: The reviewer questioned about cosine similarity as the distance measure.** Cosine similarity is better for cases where values are small. Usually errors (also known as residuals) have quite small magnitudes. We also ran comparisons using the L2 norm and didn’t find a difference in convergence rates. Our alternative metric (similarity of induced weights) also agrees with our cosine similarity metrics.

---

### Author Response · Authors · 2023-11-17
**General Response**

We thank the reviewer for providing constructive feedback. We revised our manuscript with an updated claim and experiments (updated Fig. 3 and all relevant heatmaps):

1. Transformers and Iterative Newton have the same convergence rate that is exponentially faster than Gradient Descent.

2. Gradient Descent can achieve the same-level of similarity with Transformers as Iterative Newton’s, but exponentially slower.

---

### Author Response · Authors · 2023-11-23
**General Response**

We thank again for all reviewers’ efforts. We are grateful that reviewers found our *motivation clear and promising* (eSFG), *experimental results and visualization clear* (eSFG, CBs3). We are also thankful that reviewers found our *writing clear and easy to follow* (eSFG, CBs3); and found our *experiments comprehensive* (zXY7). Finally, we are happy that reviewers (eSFG, zXY7, TsNZ)  agree *our theory is able to verify our claim*.

We first reiterate our main contribution:

1. We have shown empirically that for linear regression tasks, Transformers share the same *rate of convergence* as higher-order methods such as Iterative Newton, and both are exponentially faster than Gradient Descent.

2. We then provided theoretical results that *Transformers can implement exactly the same algorithm as Iterative Newton*. In particular, to perform $k$ Newton updates, Transformers can implement it with $\mathcal O(k)$ layers.

We then address common concerns raised by reviewers.

**The reviewers (eSFG, CBs3, zXY7) pointed out GD should converge to the OLS.** This is indeed the case, however it converges much more slowly. **We have thus updated our claims to be about convergence rates.** In our revised manuscript, we fix this issue: GD, Iterative Newton, and Transformers will all converge to the OLS solution and the major difference is the rate of convergence. GD converges exponentially slower than the other two.

**The reviewers (eSFG, CBs3, zXY7, TsNZ) asked for learning rates used for GD.** We choose the best learning rate for GD to ensure convergence, and here it’s 0.01 in well-conditioned cases. As long as GD converges, it won’t affect our claim on the rate of convergence.

**The reviewers (eSFG, TsNZ) questioned the measure of cosine similarity.** Cosine similarity is better for cases where values are small. Usually errors (also known as residuals) have quite small magnitudes. We also ran comparisons using the L2 norm and didn’t find a difference in convergence rates.

We also addressed reviewers' individual concerns and comments in separate threads.

---

### Meta-Review · Area_Chair_X9iW · 2023-12-11

**Metareview:**

This paper studies in-context learning of linear regression using transformers and argues that transformers learn something similar to the Iterative Newton's Method. The provided evidence includes theoretical constructions and empirical comparisons between Newton and Gradient Descent.

Understanding in-context learning in transformers is an important unsolved question, and a number of recent works attributed this to that transformers can implement GD. This paper challenges this view by arguing that it's more like Newton rather than GD. The reviewers found the paper generally interesting. However, they also raised concerns about the solidity of the results. During the rebuttal phase, the authors updated the main claim of the paper to be about the convergence rate difference.

**Justification For Why Not Higher Score:**

There was a significant revision of the paper in response to the reviews. The revised paper should go through another round of thorough review.

**Justification For Why Not Lower Score:**

N/A

---

### Decision · Program_Chairs · 2024-01-16

Reject